# Bacteriohopanetetrol-*x*: constraining its application as a lipid biomarker for marine anammox using the water column oxygen gradient of the Benguela upwelling system

Zoë R. van Kemenade[1], Laura Villanueva[1,2], Ellen C. Hopmans[1], Peter Kraal[1], Harry J. Witte[1], Jaap S.
Sinninghe Damsté[1,2], Darci Rush[1]

[1]Department of Marine Microbiology and Biogeochemistry, NIOZ Royal Netherlands Institute for Sea Research, Den Burg, the Netherlands
[2]Department of Earth Sciences, Geochemistry, Faculty of Geosciences, University of Utrecht, Utrecht, the Netherlands

*Correspondence to*: Zoë R. van Kemenade (zoe.van.kemenade@nioz.nl)

**Abstract.** Interpreting lipid biomarkers in the sediment archive requires a good understanding of their application and limitations in modern systems. Recently it was discovered that marine bacteria performing anaerobic ammonium oxidation (anammox), belonging to the genus *Ca.* Scalindua, uniquely synthesize a stereoisomer of bacteriohopanetetrol ('BHT-*x*'). The ratio of BHT-*x* over total bacteriohopanetetrol (BHT; ubiquitously synthesized by diverse bacteria) has been suggested as a proxy for water column anoxia. As BHT has been found in sediments over 50 Myr old, BHT-*x* has the potential to complement and extend the sedimentary biomarker record of marine anammox, conventionally constructed using ladderane lipids. Yet, little is known about the distribution of BHT-*x* in relation to the distribution of ladderanes and to the genetic evidence of *Ca.* Scalindua in modern marine systems. Here, we investigate the distribution of BHT-*x* and the application of the BHT-*x* ratio in relation to distributions of ladderane intact polar lipids (IPLs), ladderane fatty acids (FAs) and *Ca.* Scalindua 16S rRNA genes in suspended particulate matter (SPM) from the water column of the Benguela upwelling system (BUS), sampled across a large oxygen gradient. In BUS SPM, high BHT-*x* abundances were restricted to the oxygen deficient zone on the continental shelf (at $[O_2]$ <45 µmol $L^{-1}$, in all but one case). High BHT-*x* abundances co-occurred with high abundances of the *Ca.* Scalindua 16S rRNA gene (relative to the total number of bacterial 16S rRNA genes) and ladderane IPLs. At shelf stations with $[O_2]$ >50 µmol $L^{-1}$, the BHT-*x* ratio was <0.04 (in all but one case). In apparent contradiction, ladderane FAs and low abundances of BHT and BHT-*x* (resulting in BHT-*x* ratios >0.04) were also detected in oxygenated offshore waters ($[O_2]$ up to 180 µmol $L^{-1}$), whereas ladderane IPLs were undetected. The index of ladderane lipids with five cyclobutane rings ($NL_5$) correlates with *in situ* temperature. $NL_5$ derived temperatures suggested that ladderane FAs in the offshore waters were not synthesized *in situ* but were transported down-slope from warmer shelf waters. Thus, in sedimentary archives of systems with known lateral organic matter transport, such as the BUS, relative BHT and BHT-*x* abundances should be carefully considered. In such systems, a higher BHT-*x* ratio may act as a safer threshold for deoxygenation and/or *Ca.* Scalindua presence: our results and previous studies indicate that a BHT-*x* ratio of ≥0.2 is a robust threshold for oxygen-depleted waters ($[O_2]$ <50 µmol $kg^{-1}$). In our data, ratios of ≥0.2 coincided with *Ca.* Scalindua 16S rRNA genes in all samples ($n = 62$), except one. Lastly, when investigating *in situ* anammox, we highlight the importance of using ladderane IPLs over BHT-*x* and/or ladderane FAs; these latter compounds are more recalcitrant and may derive from transported fossil anammox bacteria remnants.

## 1 Introduction

Anaerobic ammonium oxidizing (anammox) bacteria are a deep branching monophyletic group belonging to the order *Planctomycetales* (Strous et al., 1999). Anammox bacteria oxidize ammonium ($NH_4^+$) to dinitrogen gas ($N_2$), using nitrite ($NO_2^-$) as an electron acceptor (Van de Graaf et al., 1995; 1997). Of the five known anammox genera, only '*Candidatus* Scalindua' has been found in open marine environments, of which the first identified species was '*Ca.* Scalindua sorokinii' in the Black Sea (Kuypers et al. 2003; Schmid et al. 2003). Since then, studies have found *Ca.* Scalindua spp. to also be present

in numerous oxygen minimum zones (OMZs) worldwide (Schmid et al., 2007; Woebken et al., 2007; Villanueva et al. 2014), where they are responsible for major losses of fixed nitrogen (*e.g.* Thamdrup et al., 2006; Schmid et al., 2007; Jensen et al., 2008; Lam et al., 2009). Although permanent OMZs (defined as $O_2 < 20$ µmol $L^{-1}$, reaching 1 µmol $L^{-1}$ in the core) constitute only ~8% of the total oceanic area (Paulmier & Ruiz-Pino, 2009), they are responsible for 20–50% of the total global nitrogen (N) loss (Gruber & Sarmiento, 1997; Codispoti et al., 2001; Gruber, 2004). Climate models predict that OMZs will expand

both spatially and temporally (*e.g.* Oschlies et al., 2018), hereby altering the biogeochemistry of the oceans. This will likely increase the potential of fixed N-loss processes, such as anammox, in marine systems (Breitburg et al. 2018).

     To constrain past and present N cycle variations, lipid biomarkers can be employed (see Rush and Sinninghe Damsté, 2017 for a review). Subsequently, biomarker information can be applied for predictions of future N cycling variations (*e.g.* Monteiro et al., 2012). Anammox bacteria uniquely synthesize ladderane fatty acids (FAs) and ladderane glycerol monoethers,

which contain three or five linearly concatenated cyclobutane rings, designated respectively as [3]- and [5]-ladderanes (Sinninghe Damsté et al., 2002, 2005; Fig. 1a). These biomarker lipids have been used to detect anammox bacteria in the natural environment (*e.g.* Kuypers et al., 2003; Hamersley et al., 2007; Jaeschke et al., 2010). Intact polar lipid (IPL)-containing ladderanes have also been used as biomarker lipids for the presence of anammox bacteria in marine waters and sediments (*e.g.* Jaeschke et al., 2010; Brandsma et al., 2011; Pitcher et al., 2011). Ladderane IPLs consist of two hydrocarbon chains, at least

one being a [3]- or [5]- ladderane, esterified or ether-bound to a glycerol moiety, which is in turn bound to a polar headgroup (phosphocholine 'PC', phosphoethanolamine 'PE' or phosphoglycerol 'PG'; Boumann et al., 2006; Rattray et al., 2008). Ladderane IPLs are thought to reflect living or recently dead anammox cells (Jaeschke et al., 2009a; Jaeschke et al., 2010; Brandsma et al., 2011; Bale et al., 2014), as they rapidly degrade after cell lysis (i.e., by headgroup cleavage; Harvey et al., 1986). By contrast, the oldest reported ladderane FAs were found in 140 kyr old sediments underlying the Arabian Sea OMZ

(Jaeschke et al. 2009b). However, partly due to the steric strain on the cyclobutane moieties, ladderane FAs degrade during diagenesis. Thus, in order to constrain the role of anammox in the N cycle during earlier time periods, a more recalcitrant biomarker is required.

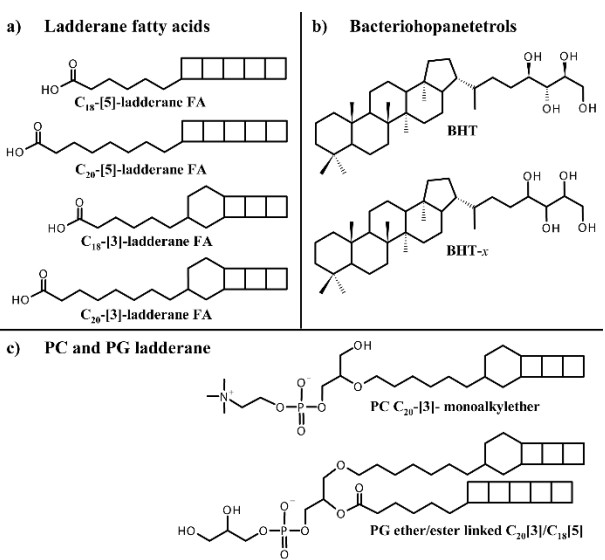

**Figure 1.** Structures of lipid biomarkers used in this study: a) ladderane fatty acids (FAs) with 5 or 3 cyclobutane moieties and 18 or 20 carbon atoms; b) Bacteriohopane-17ß, 21ß(H), 22R, 32R, 33R, 34S, 35-tetrol (BHT), ubiquitously synthesized by bacteria, and bacteriohopanetetrol stereoisomer (BHT-*x*), with an unknown stereochemistry of the tetrafunctionalised side-chain, uniquely synthesized by marine anammox bacteria; and c) Ladderane intact polar lipids PC $C_{20}$-[3]-monoalkylether ('PC ladderane') and PG ether/ester linked $C_{20}[3]/C_{18}[5]$ ('PG ladderane').

     Recently, a stereoisomer (BHT-*x*) of the ubiquitous bacteriohopanetetrol (BHT), a pentacyclic $C_{30}$ triterpenoid linked

to a tetrafunctionalised side chain, was reported to be uniquely synthesized by marine anammox bacteria, *Ca.* Scalindua (Rush et al., 2014; Schwartz-Narbonne et al., 2020; Fig. 1b). BHTs belong to the family of bacteriohopanepolyols (BHPs), which are biological precursors of hopanoids, ubiquitously found in the geological record (Ourisson & Albrecht, 1992). Intact BHPs have been found in sediments over 50 Myr old (van Dongen et al. 2006; Talbot et al. 2016; Rush et al., 2019). BHT-*x* was found to have the same distribution as ladderane FAs in sediments of Golfo Dulce, an anoxic marine enclosure in Costa Rica

(Rush et al., 2014), testifying to its potential application as anammox marker. Moreover, since BHT is ubiquitous and the

BHT-*x* stereoisomer was only found in low oxygen settings, Saénz et al (2011) had earlier proposed the ratio of BHT-*x* over total BHT (BHT-*x* ratio) to be a proxy for water column deoxygenation ([O$_2$] <50 µmol kg$^{-1}$). These studies show the potential of BHT-*x* to complement and extend the ladderane biomarker record.

To better interpret BHT-*x* as a biomarker in the sedimentary record, either as an indicator of the presence of marine anammox bacteria, *Ca.* Scalindua spp., or as a proxy for water column deoxygenation, it is imperative to establish how BHT-*x* is distributed in modern marine oxygen-depleted systems. In this study, we combine measurements of BHT-*x*, ladderane lipids (both as IPLs and FAs) and 16S rRNA marker genes in suspended particulate matter (SPM) across a redox gradient in the water column of the Benguela upwelling system (BUS). The BUS, located along the southwest African continental margin (Fig. 2), supports one of the most productive regions in the world. The high primary productivity on the broad but shallow continental shelf results in a perennial OMZ off-shelf, between ~200–500 m below sea surface (mbss). Additionally, annual variation in upwelling intensity leads to a seasonal, on-shelf oxygen deficient zone (ODZ; here defined as [O$_2$] <5 µmol L$^{-1}$ in bottom waters), which develops in late austral summer (Chapman and Shannon, 1987; Bailey et al., 1991; Mercier et al., 2003; Ekau and Verheye, 2005; Brüchert et al., 2006; Mohrholz et al., 2008). Previous research has indicated that in the BUS, anammox is responsible for major losses of bioavailable nitrogen (Kuypers et al. 2005; Kalvelage et al., 2011). The BUS therefore is an optimal modern marine system to assess the distribution of BHT-*x* in relation to that of ladderane IPLs, ladderane FAs and *Ca.* Scalindua 16S rRNA gene sequences, as a function of water column oxygenation.

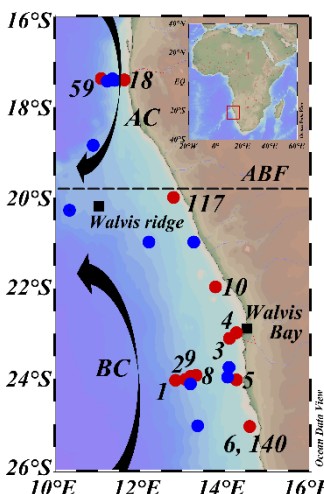

**Figure 2.** Map of sampled stations during expeditions 64PE449 (27/01/'19-14/02/'19) and 64PE450 ((15/02/'19-10/03/'19), with station dots colour coded according to sampling activities; CTD casts and nutrients (blue), and CTD casts and nutrients plus DNA and SPM sampling (red). Numbers indicate station labels of red dots. Station labels and coordinates of blue dots are given in Table S1. AC: Angolan current, BC: Benguela current, ABF: Angolan Benguela Front (~19.8°S). Maps were created in ODV using the ETOP01_2min global tiles map (Schlitzer, R, Ocean Data View, https://odv.awi.de, 2019).

## 2 Materials and methods

### 2.1 Hydrographic setting

The BUS is located off the southwest African coast, where the cold and nutrient rich waters of the Benguela current (BC) are upwelled through a combination of wind-driven Ekman transport and collision with the African continental shelf. The studied area is situated in the northern part of the BUS (16–26°S and 10–16°E; Fig. 2). Here, changes in the offshore wind field, which affect upwelling and hence primary production, result in seasonal variations and movements of the oxygen depleted waters (Chapman and Shannon, 1987). The northern border of the BUS is delineated by the dynamic Angolan Benguela Front (ABF; ~16–20°S; Fig. 2), where the warm and oligotrophic waters of the Angola current (AC), transporting the oxygen poor (<45 µmol L$^{-1}$) South Atlantic Central Water (SACW) southwards, converge with the cold and nutrient-rich waters of the equatorward BC. Seasonal variations in the intensity of the AC and the BC control dissolved oxygen (DO) concentrations in the BUS (Mohrholz et al., 1999; Brüchert et al., 2006). This results in a near permanently present OMZ located off the continental shelf (~200–500 mbss; [O$_2$] ~20–50 µmol L$^{-1}$) and a seasonally variable ODZ on the continental shelf (~50 mbss to seafloor; [O$_2$] <5 µmol L$^{-1}$), where the most severe oxygen depletion occurs during late austral summer (Chapman and

Shannon, 1987; Bailey et al., 1991; Mercier et al., 2003; Ekau and Verheye, 2005; Brüchert et al., 2006; Mohrholz et al., 2008). In the south, the northern BUS is bordered by the Lüderitz upwelling cell around ~26°S (Boyer et al., 2000).

**Table 1.** Sampling date, cruise name, station label, coordinates and sediment depth (mbss), during expeditions 64PE449 and 64PE450. Stations are grouped according to location type and listed based on their sampling date. Location type is 'shelf' when sediment depth is <120 mbss, or 'offshore' when sediment depth is >300 mbss.

| Date | Cruise | Station | Latitude (°S) | Longitude (°E) | Depth (mbss) | Location |
|------|--------|---------|---------------|----------------|--------------|----------|
| 01-02-2019 | 64PE449 | 3 | 23.096 | 14.129 | 120 | Shelf |
| 06-02-2019 | 64PE449 | 4 | 23.011 | 14.244 | 105 | Shelf |
| 07-02-2019 | 64PE449 | 5 | 24.020 | 14.303 | 100 | Shelf |
| 08-02-2019 | 64PE449 | 6* | 25.072 | 14.596 | 100 | Shelf |
| 12-02-2019 | 64PE449 | 10 | 21.968 | 13.793 | 103 | Shelf |
| 19-02-2019 | 64PE450 | 18 | 17.409 | 11.635 | 100 | Shelf |
| 01-03-2019 | 64PE450 | 117 | 20.001 | 12.823 | 105 | Shelf |
| 07-03-2019 | 64PE450 | 140* | 25.072 | 14.596 | 100 | Shelf |
| 03-02-2019 | 64PE449 | 1 | 24.056 | 12.843 | 1500 | Offshore |
| 04-02-2019 | 64PE449 | 2 | 24.042 | 13.127 | 720 | Offshore |
| 10-02-2019 | 64PE449 | 8 | 23.961 | 13.227 | 324 | Offshore |
| 11-02-2019 | 64PE449 | 9 | 23.962 | 13.226 | 407 | Offshore |
| 25-02-2019 | 64PE450 | 59 | 17.277 | 11.377 | 1000 | Offshore |

*denotes same station location, sampled 27 days apart

## 2.2 Sample collection

Between January and March 2019, two consecutive research expeditions in the Namibian BUS were undertaken with the *R/V Pelagia* (64PE449 and 64PE450). At this time, the low-oxygen waters of the SACW reached their maximum southward extension (Chapman and Shannon, 1987), which, combined with limited cross-shore bottom-water ventilation at the start of the annual upwelling cycle, led to severe oxygen depletion (Mohrholz et al., 2008). Sampling was performed at various water depths at 13 stations (11°22' 36.5"–14°47'34.8"E and 17°16'38.3"–25°12'25.0"S; Fig. 2; Table 1), covering a large range in water column oxygen concentrations (Fig. 3a; Fig. 4b, f). From here on, the deeper stations (>300 mbss stations 1, 2, 8, 9, 59), sampled in the OMZ off the continental shelf will be termed 'offshore stations', whereas the shallower stations (<120 mbss; stations 3–6, 10, 18, 117, 140) sampled on the continental shelf will be termed 'shelf stations' (Table 1). At each station, physical parameters of the water column were recorded with a Sea-Bird SBE911+ conductivity-temperature-depth (CTD) system. The CTD was equipped with an additional SBE 43 oxygen electrode (Sea-Bird Electronics, WA, USA) to measure DO (detection limit of 1–2 µmol L$^{-1}$). A NIOZ-made Rosette sampler of 24 x 12L Niskin bottles with hydraulically controlled butterfly lids was used to collect water for nutrient and DNA analysis. A ~0.5 bar overpressure of N$_2$ gas was applied to the Niskin bottles to collect water without introducing oxygen. Water samples for DNA analysis were collected into pre-cleaned acid-washed Nalgene bottles. *Ca.* 2 L of water was filtered over 0.2 µm pore diameter Millipore Sterivex filters from the Nalgene bottles kept on ice in a climate-controlled container set at 4°C. 2 mL of RNALater (RNA protect bacteria reagent, Qiagen) was applied to the Sterivex cartridges, which were then sealed with parafilm and stored at -20 °C until further processing. Suspended particulate matter (SPM) for lipid analysis was collected using four McLane Large Volume Water Transfer System Sampler (WTS-LV) in situ pumps, which were deployed for 4h (~40–900 L filtered; McLane Laboratories Inc., Falmouth, MA, USA). Water was filtered over pre-ashed GF75 grade glass fibre filters of 0.3 µm pore size and 142 mm

diameter (Advantec MFS, Inc., USA). Filters were wrapped in aluminium foil and stored at -80 °C until further processing. Water column sampling depths were chosen based on the CTD profiles, focusing around and below the redoxcline (Table S1).

## 2.3 Nutrient analysis

Samples for nutrient analysis were sub-sampled from the Niskin bottles with pre-flushed 60 mL high-density polyethylene syringes with a three-way valve. Samples for $PO_4^{3-}$ $NO_2^-$, $NO_3^-$ and $NH_4^+$ analysis were filtered over a combined 0.8–0.2 µm Supor Membrane Acrodisc PF syringe filter (PALL Corporation, NY, USA) into pre-rinsed 5 mL polyethylene vials and analysed onboard with an autoanalyzer (QuAAtro39 Gas Segmented Continuous Flow Analyser, Seal Instruments). Detection limits for $PO_4^{3-}$, $NO_2^-$, $NO_3^-$ and $NH_4^+$ were 0.005, 0.003, 0.015 and 0.019 µmol $L^{-1}$, respectively. The fixed inorganic nitrogen deficit was calculated as:

$$N\ deficit\ =\ 16\ x\ [PO_4^{3-}]\ -\ ([NO_3^-] + [NO_2^-] + [NH_4^+]) \tag{1}$$

in which 16 reflects the Redfield ratio of N:P (Redfield et al., 1963).

## 2.4. Lipid extraction and analysis

### 2.4.1 Modified Bligh and Dyer extraction

Freeze-dried filters were extracted four times using a modified Bligh and Dyer extraction (BDE) method (Bligh and Dyer, 1959; Sturt et al., 2004; Bale et al., 2021). The filters were cut in ± 1x1 cm pieces, and ultrasonically extracted (10 min) using a solvent mixture of 2:1:0.8 (*v:v:v*) methanol (MeOH), dichloromethane (DCM) and phosphate buffer, sonicated for 10 min and centrifuged for 2 min at 3000 rpm. The supernatant was then transferred to another tube, while the residue was re-extracted thrice (i.e. total of four extraction rounds), where during the last two extractions, the phosphate buffer was replaced with a trichloroacetic acid solution to enable optimal recovery of IPLs (Sturt et al., 2004). Phase separation between the solvent layer and aqueous layer was induced by adding additional DCM and phosphate buffer to obtain a ratio of 1:1:0.9 (*v:v:v*). The bottom DCM layer, containing the lipid extract, was collected, while the aqueous layer was washed two more times with DCM. The combined DCM layers were dried under $N_2$ gas. This extraction method was also performed on freeze-dried biomass from a *Ca.* Scalindua brodae enrichment culture, obtained from an anoxic sequencing batch reactor at Radboud University, Nijmegen, The Netherlands (described in Schwartz-Narbonne et al., 2019).

### 2.4.2 BHP and IPL analyses

Deuterated diacylglyceryltrimethylhomoserine (DGTS D-9; Avanti® Polar Lipids, USA) was added as an internal standard to BDE aliquots. Aliquots were then filtered over 4 mm True generated cellulose syringe filters (0.45 µm, BGB, USA) and re-dissolved in a MeOH: DCM solution of 9:1 (*v:v*) before analysis. Filtered aliquots were analysed on an Agilent 1290 Infinity I ultra high performance liquid chromatographer (UHPLC) equipped with a thermostatted auto-injector and column oven, coupled to a Q Exactive Orbitrap MS with an Ion Max source and heated electrospray ionisation probe (HESI; ThermoFisher Scientific, Waltham, MA). Separation was accomplished with an Acquity BEH $C_{18}$ column (2.1×150 mm, 1.7 µm, Waters) maintained at 30°C. An eluant of (A) MeOH∕$H_2O$∕formic acid∕14.8 M $NH_3$aq with a ratio of 85:15:0.12:0.04 (*v:v:v:v*) and (B) IPA∕MeOH∕formic acid∕14.8 M $NH_3$aq with a ratio of 50:50:0.12:0.04 (*v:v:v:v*) was used, with a flow rate of 0.2 mL $min^{-1}$. The elution program was set at: 5% B for 5 min, followed by a linear gradient to 40% B at 12 min and then to 100% B at 50 min, which was maintained until 80 min. Positive ion HESI settings were: capillary temperature, 300°C; sheath gas ($N_2$) pressure, 40 arbitrary units (AU); auxiliary gas ($N_2$) pressure, 10 AU; spray voltage, 4.5 kV; probe heater temperature, 50°C; S-lens 70 V. Lipids were detected using positive ion monitoring of *m/z* 350–2000 (resolution 70,000 ppm at *m/z* 200), followed

by data dependent MS$^2$ (resolution 17,500 ppm) of the ten most abundant ions and dynamic exclusion (for 6 s) within 3 ppm mass tolerance. An inclusion list with the calculated exact masses of 165 calculated BHPs was applied. Optimal fragmentation was achieved with a stepped normalized collision energy of 22.5 and 40 (isolation width 1.0 *m/z*) for BHP analysis (Hopmans et al., 2021) and 15, 22.5 and 30 (isolation width, 1.0 *m/z*) for IPL analysis (Bale et al., 2021). The Q Exactive Orbitrap was calibrated every 48 h using the ThermoScientific Pierce LTQ Velos ESI Positive Ion Calibration Solution. The summed mass chromatograms of the ammoniated ([M+NH$_4$]$^+$; *m/z* 564.499) and sodiated ([M+Na]$^+$; *m/z* 569.454) adducts of BHT(-*x*) were integrated within 3 ppm mass accuracy. The BHT-*x* ratio was then calculated as:

$$BHT - x\ ratio\ =\ \frac{BHT-x}{(BHT+BHT-x)} \tag{2}$$

All ladderane IPLs reported by Rattray et al. (2008) were evaluated in the BUS SPM samples and a *Ca.* Scalindua brodae enrichment culture (see Table S4 for exact masses). Only PC C$_{20}$[3] monoalkylether (from here on termed 'PC ladderane) and an ether-ester PG C$_{20}$[3]-C$_{18}$[5] (from here on termed 'PG ladderane') were detected in the BUS SPM samples (Fig. 1c). For the PC ladderane, the exact ion mass was used for integration ([M]$^+$; *m/z* 530.361). For the PG ladderane, the combined protonated ([M+H]$^+$; *m/z* 775.491) and ammoniated ([M+NH$_4$]$^+$; *m/z* 792.517) adduct was used for integration (within 3 ppm mass accuracy). BDE of the *Ca.* S. brodae enrichment culture (containing both ladderane IPLs and BHT-*x*) was used as a quality control sample in each sequence run. Due to a lack of commercially available quantification standards for BHT(-*x*) and ladderane IPLs, abundances are reported as the peak area response (response unit, ru) per litre of filtered water. Although this does not allow for quantification of absolute concentrations, it does allow for quantification of the relative abundances, as the response factor should be identical across the sample set.

### 2.4.3 Ladderane fatty acids

SPM from stations 2, 6 and 140 were additionally analysed for ladderane FAs. BDE aliquots were saponified by adding 2 ml of KOH (1 M in 96% MeoH) and refluxing for 1 h. Then, 2 mL of bidistilled water was added and the pH was adjusted to 3 with 1:1 HCL:MeOH (*v:v*). To collect the fatty acid fraction, 2 mL of DCM was added, after which the tube was sonicated for 5 min and centrifuged for 2 min at 3000 rpm. The fatty acid fraction (DCM layer) was collected, and the procedure was repeated two more times. Fatty acid fractions were then dried over a sodium sulfate (Na$_2$SO$_4$) column. The fractions were then methylated using diazomethane to convert FAs into their corresponding fatty acid methyl esters (FAMEs). To remove polyunsaturated fatty acids (PUFAs), extracts were dissolved in DCM and eluted over a silver nitrate (AgNO$_3$) impregnated silica column. Lastly, the FAME fractions were filtered through a 0.45 µm PTFE filter (BGB, USA) using acetone.

### 2.4.4 Ladderane fatty acid analysis

Purified FAME fractions were analysed on an Agilent 1290 Infinity I ultra high performance liquid chromatographer (UHPLC) equipped with a thermostatted auto-injector and column oven, coupled to a Q Exactive Plus Orbitrap MS, with an atmospheric pressure chemical ionization (APCI) probe (Thermo Fischer Scientific, Waltham, MA). Separation was realised with a ZORBAX Eclipse XDB C$_{18}$ column (Agilent, 3.0×250 mm, 5 µm), maintained at 30°C. MeOH was used as an eluant at 0.18 ml min$^{-1}$ with a total run time of 20 min. Optimal APCI source settings were determined using a qualitative standard mixture of [3]- and [5]-ladderane FAMEs. Positive ion APCI source settings were: corona discharge current, 2.5 µA; source CID, 10 eV; vaporizer temperature, 475°C; sheath gas flow rate, 50 arbitrary units (AU); auxiliary gas flow rate, 30 AU; capillary temperature, 300°C; S-lens, 50 V. A mass range of *m/z* 225–380 was monitored (resolution 140,000 ppm), followed by a data-dependent MS$^2$ (resolution 17,500 ppm at *m/z* 200), in which the ten most abundant masses in the mass spectrum were fragmented successively (stepped normalized collision energy 20, 25, 30). An inclusion list containing the exact masses of

C$_{14-24}$ [3]- and [5] ladderane FAMEs was used. Mass chromatograms (within 5 ppm mass accuracy) of the protonated molecules ([M+H]$^+$) were used to integrate C$_{18}$[3]-, C$_{18}$[5]-, C$_{20}$[3]- and C$_{20}$[5]-ladderane FAMEs (*m/z* 291.232, 289.216, 319.263, 317.248, respectively). Ladderane FAMEs were quantified by external calibration curves of isolated standards of the C$_{20}$[3]-

and [5] ladderane FAMEs (Hopmans et al., 2006; Rattray et al., 2008; Rush et al., 2011). Ladderane FAME standards were isolated from an anammox enrichment culture grown in sequencing batch reactors, containing both *Ca.* Scalindua wagneri and *Ca.* Kuenenia stuttgartiensis (described in Kartal et al., 2006). The index of ladderane lipids with five cyclobutane rings (NL$_5$) was calculated to quantify the trends in ladderane chain lengths with respect to temperature, using:

$$NL_5 = \frac{C_{20}[5]ladderane\ FA}{(C_{18}[5]ladderane\ FA + C_{20}[5]ladderane\ FA)} \tag{3}$$

Following, the relationship between NL$_5$ and temperature is then given by:

$$NL_5 = 0.2 + \frac{0.7}{(1 + e^{-\left(\frac{T-16.3}{1.5}\right)})} \tag{4}$$

with temperature (T) in °C (Rattray et al., 2010).

**2.5 DNA extraction and phylogenetic analysis**

Millipore® Sterivex™ filters were extracted for DNA using the Qiagen DNeasy Powersoil kit®. PCR reactions of the DNA templates were conducted with the Qiagen® PCR reagents (Taq PCR Master Mix Kit). The universal prokaryotic primer pair, forward 515F-Y and reverse 926R (Parada et al. 2016) was used to target the V4–V5 small subunit ribosomal RNA region,

and was modified with 12 different nucleotide barcodes at both the forward and reverse primer. The 515F-Y/926R primer pair has been found to successfully target *Ca.* Scalindua (*e.g.* Yang et al., 2020) and is reported to reflect marine community compositions well (Parada et al., 2016). Reagents were mixed with fifty times diluted DNA template (2 µL) by addition of: 11.75 µL Nuclease free water, 5 µL 5x Qiagen Phusion buffer, 2 µL dNTPs (2.5 mM), 3 µL of 515F-Y/926R primer pair (4 µmol L$^{-1}$), 1 µL BSA (20 mg ml$^{-1}$) and 0.25 µL Taq polymerase (5 units µL$^{-1}$). Negative controls were included during

extractions and PCR reactions. Amplification was performed using the following PCR program: 30 s at 98°C (1 cycle), 10 s at 98° followed by 20 s at 50°C and 30 s at 72°C (30 cycles), 7 min at 72°C (1 cycle) and ending with 5 min at 4°C. To quantify DNA concentrations of the PCR reagents, PCR products were mixed with a Xylene Ficoll loading dye and loaded on a 2% agarose gel, together with a home-made *Escherichia coli* quantification standard dilution series (20, 10 and 1 ng µL$^{-1}$). Gel electrophoresis was performed for 1 h at 75 V. After, the gel was stained with Ethidium bromide. Gels were imaged using

GeneSys (lightning, TLUM – Mid Wave; filter, UVO32). The 400 bp bands were then quantified using the 'QuickQuant' option. Following quantification, all PCR products were pooled in equimolar amounts (40 ng DNA per sample) and loaded on a 2% gel stained with SYBRsafe®. The 400 bp band was extracted from the gel using the QIAquick® PCR Gel Extraction Kit. Concentration of the pooled PCR product (20 ng uL$^{-1}$) was quantified using Qbit (Thermo Fisher Scientific Inc.). Library preparation was performed with a 16S V4–V5 library preparation kit and sequencing with an Illumina MiSeq 2x300 bp

sequencing platform (Illumina, San Diega, CA) at the University of Utrecht Sequencing Facility (USEQ, the Netherlands).

The prokaryotic 16S rRNA gene amplicon sequences were analysed using the *Cascabel* data analysis pipeline (Abdala Asbun et al. 2020). Raw forward and reverse reads were merged using PEAR (Zhang et al. 2014). Barcode reads were demultiplexed using QIIME (Caporaso et al. 2010), allowing a maximum of 2 barcode mismatches, a maximum of 5 consecutive low-quality base calls and a maximum unaccepted Phred quality score of 19. Reads were then filtered based on

length using the values of the median distribution, with an offset of 10 bp. Sequences were dereplicated with VSEARCH, and subsequently clustered to operational taxonomic units (OTUs) using the UCLUST algorithm in QIIME, with a 97% threshold. From each OTU, the most abundant sequence was picked as representative with QIIME (Caporaso et al, 2010). Taxonomy

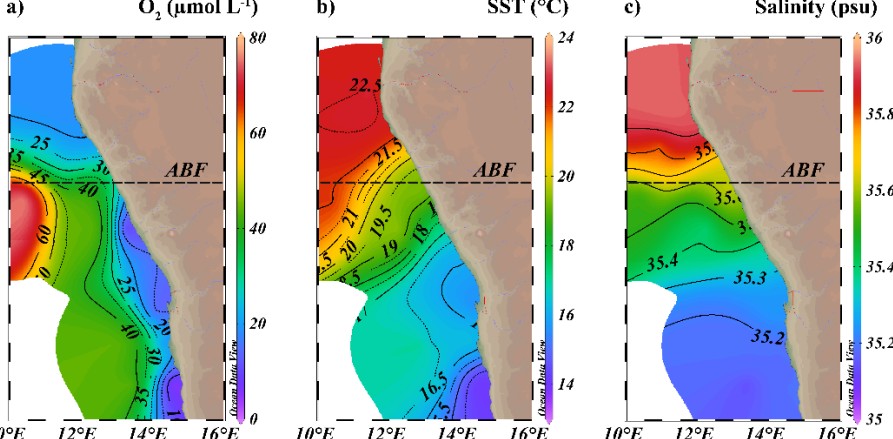

**Figure 3.** Isosurface plots of a) minimum observed oxygen concentration in water column (µmol L$^{-1}$), b) sea surface temperature (SST; °C) and c) maximum observed salinity in water column (psu). ABF: Angolan Benguela Front (~19.8°S). Maps were created in ODV using the ETOP01_2min global tiles map 150x75 weighted average gridding (Schlitzer, R, Ocean Data View, https://odv.awi.de, 2019).

**Longitudinal section plots**

**Latitudinal section plots**

**Figure 4.** Longitudinal (a-d) and latitudinal (e-h) section plots showing interpolated concentrations of (a,e) BHT-*x* (ru L$^{-1}$), (b,f) dissolved oxygen (µmol L$^{-1}$), (c, g) ladderane IPLs (ru L$^{-1}$) and (d ,h) nitrogen deficit (µmol L$^{-1}$). Contour lines for O$_2$ and N deficit concentrations are shown.

was assigned based on the SILVA database (SSU 138 Ref NR 99), using the VSEARCH tool. OTUs representative of the order Brocadiales (to which *Ca.* Scalindua spp. belong) were extracted with QIIME using filter_taxa_from_otu_table.py. The filtered sequences were imported in the SILVA NR99 SSU 138 Ref database using the ARB parsimony tool in the ARB software package (ARB SILVA, Germany) to assess the phylogenetic affiliation of the partial 16S rRNA gene sequences. Affiliated sequences were checked for homology and imported in MEGAX using the BLAST search query (Kumar et al. 2018). The twelve OTUs with the largest number of reads and 27 reference sequences were aligned, based on 422 bp, with the Clustal W alignment tool. A Kimura 2-parameter model with *Gamma* distributed sites was then used to calculate pairwise distances between sequences and to create a maximum likelihood tree in MEGAX, using a bootstrap with 1,000 replicates and the maximum parsimony method to create the initial tree (Kimura, 1980; Kumar et al. 2018). To estimate the relative abundance of *Ca.* Scalindua spp. 16S rRNA reads in relation to the total amount of bacterial 16S rRNA reads, relative *Ca.* Scalindua spp. reads were calculated for each sample (in % of total bacterial reads). Though this does not allow for absolute quantification, it does allow for a comparison of relative abundances throughout the dataset of this study, as all samples were processed and analysed in the same way.

## 2.6 Statistical analysis

A multivariate binomial regression was performed with anammox biomarker lipids and *Ca.* Scalindua 16S rRNA gene amplicon sequences. 16S rRNA gene amplicon sequences were used dichotomously, defined as either the presence or absence of *Ca.* Scalindua assigned OTUs in a given sample. Pearson's correlations between anammox lipid biomarkers and the physiochemical parameters were also investigated (Matlab, R2019a).

## 3 Results

### 3.1 Hydrography and seawater chemistry

The location of the Angolan Benguela front (ABF) during the expedition was ~19.8°S, close to Walvis Ridge, where large horizontal gradients in sea surface temperature (SST; temperatures integrated between 0–30 mbss) and salinity (measured at the halocline) were observed, and surface isotherms and isohalines fanned out seawards (Fig. 3b and c). At the time of sampling, upwelled waters from the BC were clearly distinguishable at the stations south of the ABF (stations 1–6, 10, 117 and 140; 20°S–26°S), where SST and salinity are relatively low, ranging between 12.9–19.3°C and 35.1–35.6 psu, respectively (Table S1). Offshore (bottom depth ~300–1500 mbss), a weak OMZ was present between ~100–500 mbss, with $[O_2]$ levels down to ~40 µmol $L^{-1}$. In the shallow waters on the continental shelf (bottom depth <120 mbss) an ODZ was present, with $[O_2]$ decreasing rapidly with depth from 30 mbss onwards, with nearly anoxic waters ($[O_2]$ ~1.5–5.5 µmol $L^{-1}$) below ~50–80 mbss. The strongest oxygen depletion was found at the shallowest stations (<100 mbss; stations 4, 5, 6, 10, 117 and 140), while at the slightly deeper shelf station 3 (~120 mbss), $[O_2]$ in bottom waters was not lower than ~20 µmol $L^{-1}$ (Fig. 4b; Fig. 5c). The influence of the AC was apparent at stations north of the ABF (stations 18 and 59), where SST and salinity are relatively high (Fig. 3b and c), ranging between 16.9–22.9°C and 35.7–35.9 psu, respectively (Table S1). In these waters, $[O_2]$ declined to ~20 µmol $L^{-1}$ between ~50–500 mbss (Fig. 4f; 5j). During the expeditions, water column nutrient concentrations in the BUS ranged between: 0.0–4.9 µmol $L^{-1}$ for $NO_2^-$, 0–40 µmol $L^{-1}$ for $NO_3^-$, and 0.1–9.8 µmol $L^{-1}$ for $NH_4^+$ (Fig. 5d–f, k–m; Table S2). Nearly all sampled stations exhibited an N deficit (eq. 1) throughout the water column, with the strongest deficit being observed at the shelf stations (Fig. 4d, h; Table S2). The N deficit was highest at station 6, reaching a deficit of ~49 µmol $L^{-1}$ at 85 mbss. When station 6 was resampled in March (station 140), the deficit had decreased to a maximum of ~23 µmol $L^{-1}$ at 10 mbss. The only station in the BUS where no N deficit was observed was station 117.

## 3.2 Anammox lipid biomarkers

### 3.2.1 BHT and BHT-*x* abundances and ratio

BHT was found in all SPM samples, except one (station 117, 40 mbss). BHT abundance ranged between $1.0 \times 10^5$–$2.1 \times 10^8$ ru L$^{-1}$ at the shelf stations and between $1.1 \times 10^5$–$1.8 \times 10^7$ ru L$^{-1}$ at the offshore stations (Table S3). BHT-*x* was found exclusively in the southern part of the northern BUS. No BHT-*x* was observed at stations located near the ABF (station 117) nor north of the ABF (stations 18 and 59). When present, the BHT-*x* abundance ranged from $1.4 \times 10^5$–$1.6 \times 10^7$ ru L$^{-1}$ (Fig. 4a, e; Fig. 5a, h; Table S3). At the southern stations located on the shelf (stations 3–6, 10 and 140), BHT-*x* was observed below ~30 mbss. At these stations, the BHT-*x* abundance ranged between $2.3 \times 10^5$–$1.6 \times 10^7$ ru L$^{-1}$. At the southernmost shelf station, when sampled in February (station 6), BHT-*x* was observed between 50–85 mbss, increasing in abundance from $8.5 \times 10^5$ ru L$^{-1}$ at 50 mbss to $8.3 \times 10^6$ ru L$^{-1}$ at 85 mbss (Fig. 5a). In March (station 140), BHT-*x* was present at all sampled depths (35–85mbss), ranging between $1.5$–$5.3 \times 10^6$ ru L$^{-1}$ (Fig. 5a). At the southern offshore stations (stations 1, 2, 8, and 9), BHT-*x* was present between ~200–400 mbss and in the bottom waters of station 1 (1500 mbss) and 2 (710 mbss). Here, the abundance ranged between $1.4 \times 10^5$–$1.2 \times 10^6$ ru L$^{-1}$ (Fig. 5h). The BHT-*x* ratio (eq. 2) in the BUS water column ranged between 0.00–0.55 (Fig. 6, Table S3).

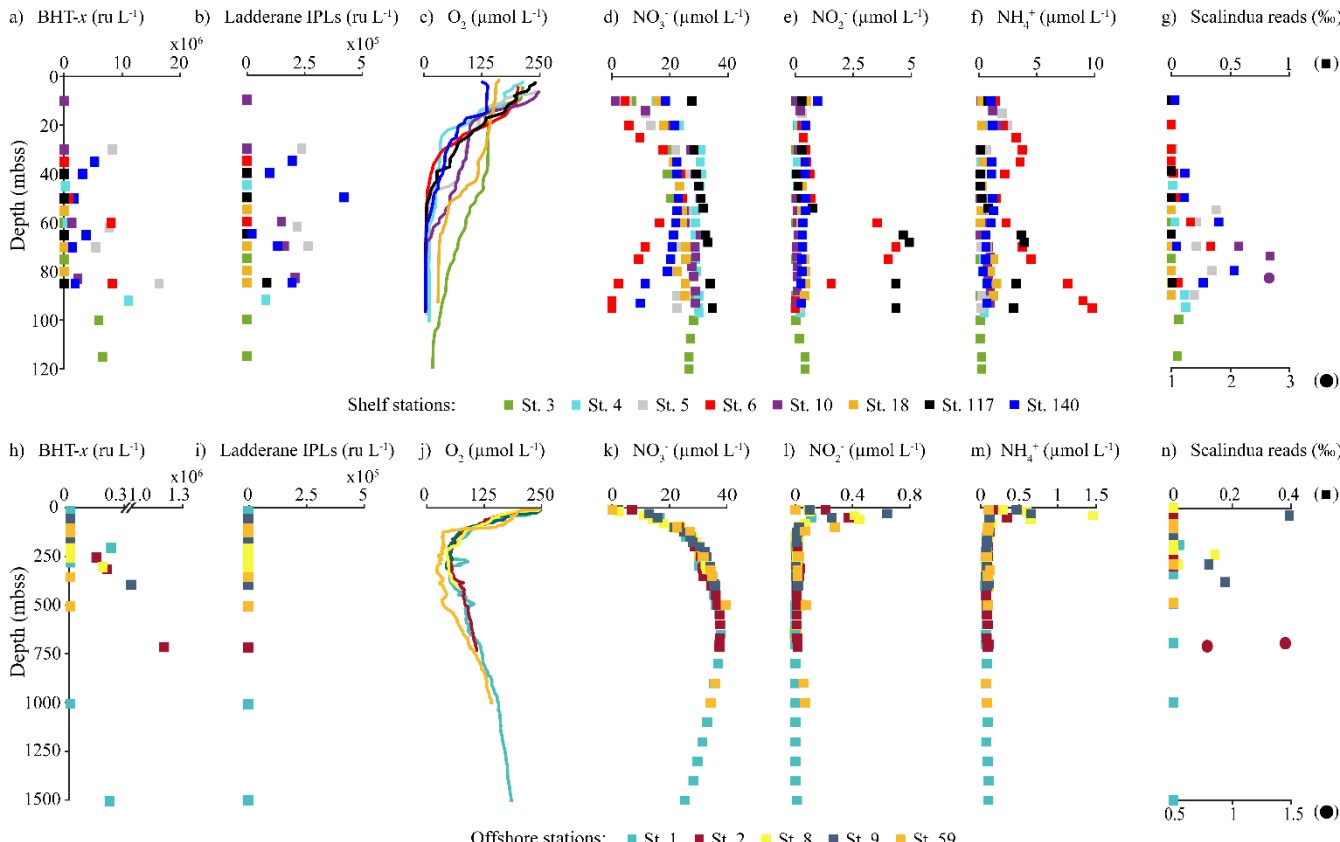

**Figure 5.** Concentration and relative abundance profiles in the water column of the BUS at shelf (top panels) and offshore (bottom panels) stations. (a, h) BHT-*x* abundance (ru L$^{-1}$), (b, i) ladderane IPL abundance (ru L$^{-1}$), concentrations in µmol L$^{-1}$ of (c, j) oxygen [O$_2$], (d, k) nitrate [NO$_3^-$], (e, l) nitrite [NO$_2^-$], (f, m) ammonium [NH$_4^+$] and relative abundance of (g, n) *Ca.* Scalindua spp. reads in permille of total bacterial reads (NB circles are plotted on bottom x-axis with a different scale).

### 3.2.2 Ladderane IPLs

The presence of all ladderane IPLs reported for the *Ca.* Scalindua brodae enrichment culture (Table S4) and those previously reported for *Ca.* Scalindua spp. (Rattray et al., 2008) was evaluated in the BUS SPM samples. However, at the time of sampling, only the PC and PG ladderanes (Fig. 1c) were detected in the BUS water column. Furthermore, these ladderane IPLs were found in SPM from a limited number of shelf stations located south of the ABF (stations 5, 10, 117 and 140), below~30

mbss waters. Concentrations ranged between $1.1 \times 10^4 – 6.4 \times 10^5$ ru L$^{-1}$ for the PG ladderane and between $2.1 \times 10^4 – 4.2 \times 10^5$ ru L$^{-1}$ for the PC ladderane (Table S3). At station 5, PC and PG ladderanes were both present. At stations 10, 117 and 140 the PG ladderane was less abundant or absent at the water column depths where the PC ladderane was found. From here on, their summed abundances are reported as 'ladderane IPLs' (Fig. 4c, g; Fig. 5b, i).

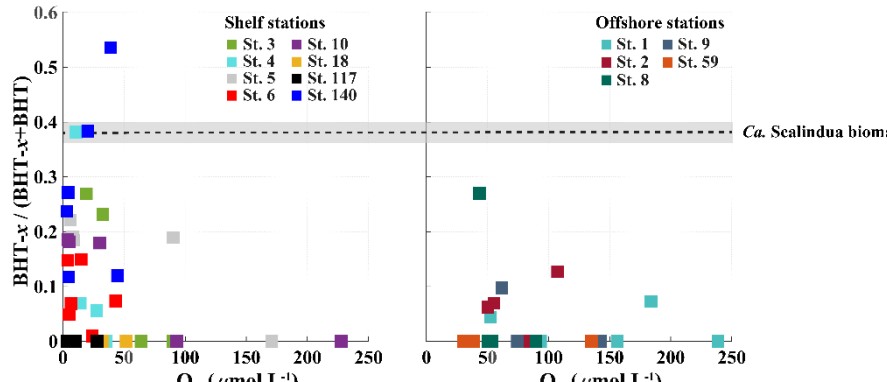

**Figure 6.** Scatter plots showing relationship between BHT-$x$ ratio (BHT-$x$/(BHT+BHT-$x$) and oxygen concentration (µmol L$^{-1}$) for stations located on the shelf (left) and stations located offshore (right). The BHT-$x$ ratio observed in the *Ca.* S. brodae enrichment culture is indicated with the black dotted line at 0.38. The grey area corresponds to the standard deviation of the ratio (±0.02) between different HPLC/MS runs.

### 3.2.3 Ladderane FAs

Ladderane FAs (i.e. $C_{18}[3]$-, $C_{18}[5]$-, $C_{20}[3]$- and $C_{20}[5]$-ladderane FAs) were analysed as FAMEs in the SPM from an offshore
station (station 2) and of the southernmost shelf station, sampled in February (station 6) and in March (station 140). Summed ladderane FA concentrations ranged between 15–320 pg L$^{-1}$ (Fig. 7; Table S5). At offshore station 2 (Fig 7a), of the six sampled depths, ladderane FAs were observed at 125, 250 and 710 mbss, with peak concentrations at 250 mbss (320 pg L$^{-1}$). At the southernmost shelf station sampled in February (station 6; Fig 7b), highest ladderane FA concentrations were observed at the deepest sampling depth (85 mbss, 140 pg L$^{-1}$). Concentrations an order of magnitude lower were found at the oxycline (30
mbss; 20 pg L$^{-1}$) and at 60 mbss (30 pg L$^{-1}$), while no ladderane FAs were observed at 35 or 40 mbss. At this location in March (station 140; Fig 7c), ladderane FAs were present throughout the oxygen-depleted interval of the water column (35–85 mbss), with total concentrations ranging between 20–60 pg L$^{-1}$. The NL$_5$ index was calculated for all SPM samples that contained both $C_{20}[5]$ and $C_{18}[5]$ FAs (Table S5). For station 2, the NL$_5$ was 0.9 at 250 mbss and 0.5 at 710 mbss. For station 6 and 140, the NL$_5$ ranged between 0.2–0.4. No evidence was found for the presence of short chain biodegradation products (i.e. $C_{14}[3]$-
, $C_{14}[5]$-, $C_{16}[3]$-, $C_{16}[5]$- ladderane FAs; Rush et al., 2011), nor $C_{22-24}$-ladderane FAs.

### 3.3 16S rRNA gene analysis

16S rRNA gene amplicon sequencing (422 bp fragment) analysis was conducted on SPM collected at 13 stations from various water column depths, including the same or similar depths as analysed for lipids. A total of 10,283,136 bacterial reads were recovered from the BUS stations (excluding singletons), of which 0.14‰ could be taxonomically assigned to the genus *Ca.*
Scalindua (Arb SILVA 132R database). These sequences were further analysed to assess the distribution and phylogeny of *Ca.* Scalindua in the BUS. Negative controls did not contain reads taxonomically assigned to *Ca.* Scalindua spp.

### 3.3.1 Distribution *Ca.* Scalindua spp. 16S rRNA sequences in the BUS

The relative abundances of *Ca.* Scalindua spp. 16S rRNA gene sequences in respect to the total amount of bacterial 16S rRNA gene sequences was calculated to estimate the distribution of *Ca.* Scalindua in the BUS (Figs. 5g, n). Sequences taxonomically
assigned to *Ca.* Scalindua spp. were detected in 11 out of 13 stations, in SPM collected at the shelf (<120 mbss; stations 3–6, 10 and 140) and at offshore stations (>300 mbss; stations 1, 2, 8 and 9) but were not detected at stations located north of the ABF (stations 18 and 59). At shelf stations (Fig. 5g), the relative *Ca.* Scalindua spp. gene read abundance ranged between 0–2.7‰, with highest abundances found below 50 mbss. In surface waters (<30 mbss), no *Ca.* Scalindua spp. 16S rRNA gene

sequences were detected, except at station 140, where *Ca.* Scalindua spp. was present throughout the water column (0.03–0.53‰). At offshore stations, the relative abundance of *Ca.* Scalindua spp. 16S rRNA gene copies ranged between 0–1.5‰, with highest relative abundances found in bottom waters at station 2 at 700 and 710 mbss (Fig. 5n). At offshore stations 1, 8 and 9, *Ca.* Scalindua spp. was detected between 50–390 mbss.

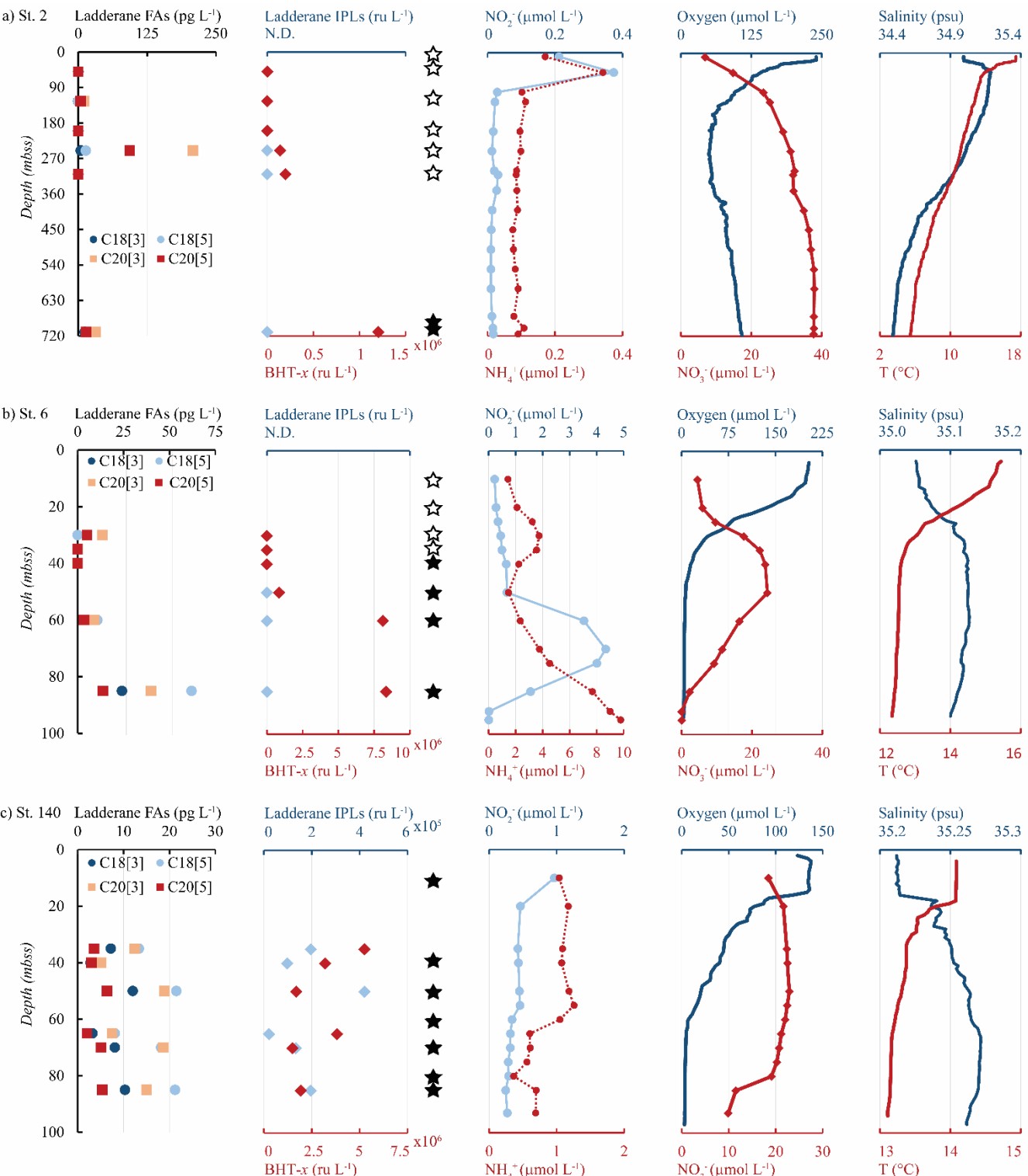

**Figure 7.** Water column depth profiles of stations (a) 2, (b) 6, and (c) 140, showing distributions of (from left to right) ladderane FAs (pg L⁻¹), ladderane IPLs and BHT-*x* (ru L⁻¹), nitrite and ammonium concentrations (µmol L⁻¹), oxygen and nitrate concentrations (µmol L⁻¹) and temperature (°C) and salinity (psu). Star symbols indicate water column depths where SPM was collected for DNA analysis. Filled stars represent depths where *Ca.* Scalindua spp. 16S rRNA gene sequences were detected. Stations 6 and 140 were sampled at the same location 27 days apart (station 6 in February and station 140 in March).

### 3.3.2 *Ca.* **Scalindua phylogeny**

*Ca.* Scalindua 16S rRNA reads (422 bp) were assigned to 66 operational taxonomic units (OTU 1–66) based on 97% sequence similarity (Table S6). Most of the *Ca.* Scalindua spp. reads (88%) could be assigned to twelve OTUs (OTU-1 to -12) ranging from the relative most abundant to least abundant OTU (Fig. S1). To estimate the phylogenetic relationship of these 12 OTUs to *Ca.* Scalindua spp. sequences from other OMZs, a maximum likelihood tree was constructed with reference sequences from various other OMZs and anammox enrichment cultures (Fig. 8). In addition, the pairwise distances between these sequences (based on 422 bp) were calculated (Table S7). The phylogeny of the BUS OTUs revealed a cluster of ten OTUs (OTUs 1–4 and 6–9, 11 and 12; Fig. 8), with an overall sequence identity of 96%. OTU-10 displays a relatively large evolutionary divergence (>12%; Fig. 8) and limited sequence identity (88%) to the other BUS OTUs. Highest sequence identity of the BUS OTU cluster is observed with environmental sequences isolated from the Guaymas deep sea hydrothermal vent sediment (98%) and the Black Sea suboxic waters (53 mbss; also 98%), which in turn both exhibited the highest sequence identity to *Ca.* Scalindua sorokiini (again 98% in both cases). Sequence identity to *Ca.* Scalindua brodae and *Ca.* Scalindua spp. sequences detected previously in the Namibian OMZ (Kuypers et al., 2005) was 96%. Lowest sequence identity is seen with *Ca.* Scalindua wagneri (93%) and the Arabian Sea (94%). OTU-5 is placed outside of the BUS cluster (Fig. 8) and shows the highest sequence identity to Gulf of Mexico and Indian Ocean sediments (98% and 97% respectively). Sequence identity of OTU-5 in relation to the BUS OTU cluster is 94%.

### 4 Discussion

In the BUS, seasonal shifts in upwelling intensity create large spatiotemporal variability in oxygen concentrations (Bailey, 1991). Anammox has been reported previously in the low oxygen BUS water column (Kuypers et al. 2005), which therefore presents an ideal location to investigate anammox biomarkers. We assessed the distribution of BHT-*x* across the redox gradient in the BUS, to provide further insights into the application of BHT-*x* as a biomarker for *Ca.* Scalindua spp., and its ratio over total BHT (BHT-*x* ratio) as a proxy for deoxygenation in dynamic upwelling systems.

### 4.1 Spatiotemporal distribution of anammox biomarkers along the redox gradient in the BUS

### 4.1.1 Anammox markers are constrained to the ODZ in the southern BUS shelf waters

During expeditions 64PE449 and 64PE450, the seasonal ODZ had developed in the southern BUS shelf waters (20°S–26°S) between ~50 mbss and the seafloor, with $[O_2]$ down to ~1.5–5.5 µmol $L^{-1}$. Nutrient analyses revealed a large N deficit in the ODZ (N deficit >10 µmol $L^{-1}$; Fig. 4d), suggesting major losses of bioavailable N by anammox and/or denitrification. Relatively high BHT-*x* abundances are detected at the southern shelf stations (stations 3–6, 10, 140) below 30 mbss (Fig. 5a). 16S rRNA gene analysis indeed indicated on-shelf presence of *Ca.* Scalindua spp. in the water masses below 30 mbss, with the highest abundance of *Ca.* Scalindua spp. 16S rRNA gene sequences relative to the total bacterial 16S rRNA gene sequence pool below ~50 mbss (Fig. 5g). In addition to the *Ca.* Scalindua spp. 16S rRNA gene, ladderane IPLs are thought to reflect living or recently dead anammox cells (Harvey et al., 1986; Jaeschke et al., 2009a; Brandsma et al., 2011; Bale et al., 2014). Ladderane IPLs are found on-shelf (stations 4, 5, 10, 117, and 140) between 30 mbss to the seafloor. The N deficit was significantly correlated with both BHT-*x* and ladderane IPLs (r(60) = 0.53, $\rho$ = <0.001; Table 2) and on-shelf N deficiencies were accompanied with relatively high BHT-*x* and ladderane IPL abundances (Fig. 4). This suggests anammox was at least in part responsible for loss of bioavailable N. In summary, the co-occurrence of BHT-*x* with *Ca.* Scalindua spp. 16S rRNA reads, ladderane IPLs and on-shelf N deficiencies, indicate the presence of living (or recently dead) anammox cells in the BUS shelf waters (below ~30 to 50 mbss), consistent with earlier reports of anammox activity on the Namibian continental shelf waters (Kuypers et al., 2005).

Additionally, *Ca.* Scalindua spp. 16S rRNA gene sequences and anammox biomarkers were detected in the more oxygenated surface shelf waters (above 50 mbss), at ambient $[O_2]$ up to ~45 µmol $L^{-1}$ (and up to ~90 µmol $L^{-1}$ in one case), surpassing earlier established oxygen limits for anammox. Culturing studies have indicated that anammox bacteria are already inhibited at $[O_2]$ as low as 1 µmol $L^{-1}$ (Strous et al., 1997). In the environment, *Ca.* Scalindua spp. has been shown to remain active at $[O_2]$ up to 9–20 µmol $L^{-1}$ in the Namibian and Peruvian OMZs respectively (Kuypers et al., 2005; Hamersley et al.,

2007; Kalvelage et al., 2011) and up to ~9 µmol $L^{-1}$ in the Black Sea (Jensen et al., 2008). A possible explanation has been provided by Woebken et al. (2007), who showed that *Ca.* Scalindua spp. colonize microscopic particles in the BUS, which provide suitable anaerobic micro-niches. Nevertheless, this was found to be restricted to ambient $[O_2]$ levels below 25 µmol $L^{-1}$. Likely, our evidence for the presence of anammox bacteria in the more oxygenated BUS shelf waters reflects material transported upwards from the deeper ODZ. Upwelled waters from the BC were clearly distinguishable at stations south of the

ABF (stations 1–6, 10, 117 and 140; 20°S–26°S), as indicated by the relatively low SST and salinity at the halocline (Fig. 3b, c; Table S1).

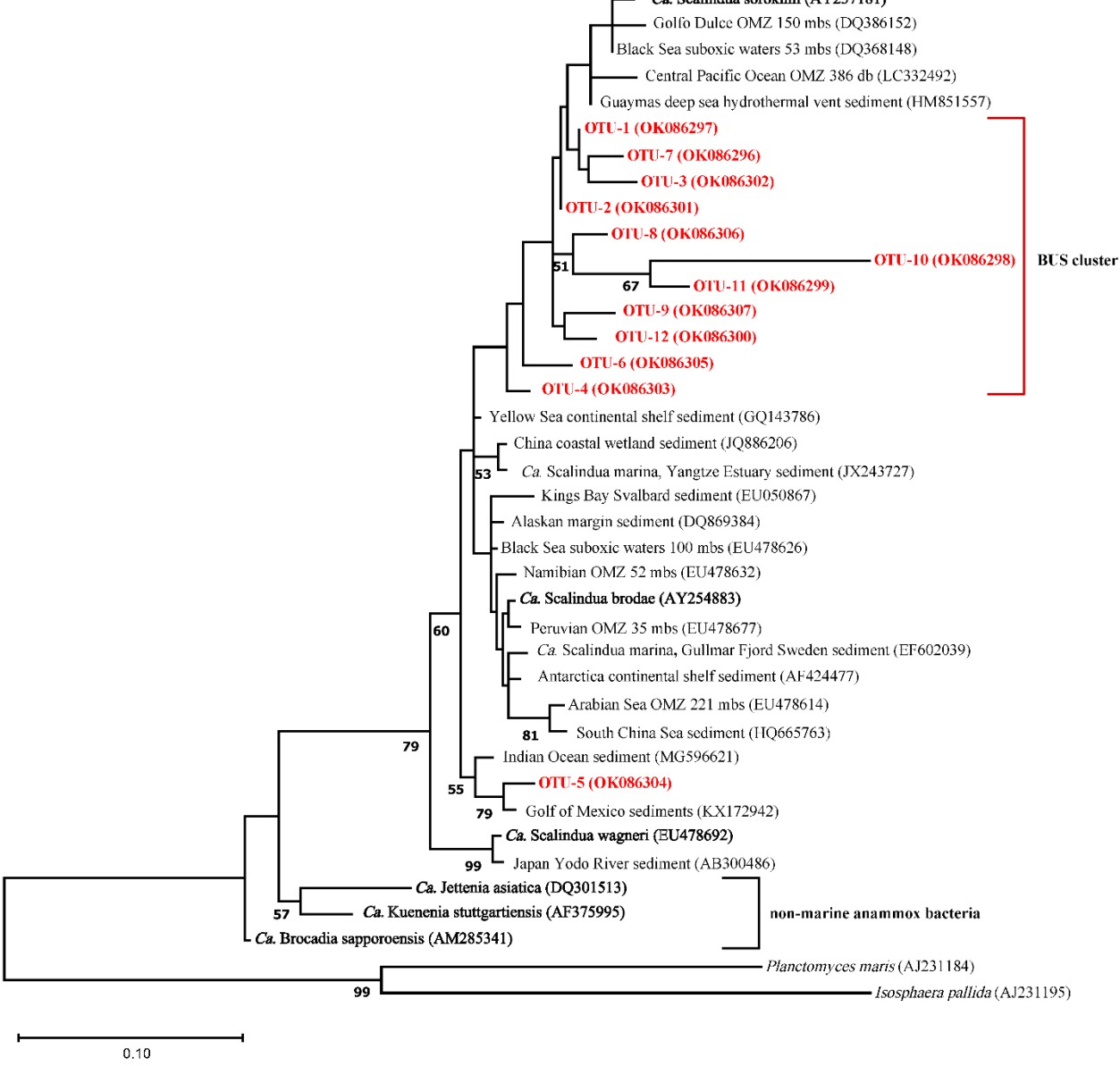

**Figure 8.** Maximum likelihood phylogenetic tree based on partial (422 bp) 16S rRNA gene sequences displaying the relationship of *Ca.* Scalindua spp. sequences of the 12 most abundant OTUs detected in the BUS (in red/bold; see Fig. S1 for relative abundance of each OTU)
with other sequences from marine OMZs and sediments (in black), and anammox bioreactors (in black/bold). Bootstrap values higher than 50% are indicated in the nodes. The scale bar represents 10% estimated sequence divergence. The outgroup is formed by 16S rRNA gene sequences of *Planctomyces maris* and *Isosphaera pallida*. NBCI accession numbers are indicated in between parentheses.

## 4.1.2 Absence of anammox biomarkers near and north of the Angolan Benguela Front

At the end of austral summer (i.e. the timing of expeditions 64PE449 and 64PE450), the ABF reaches its most southern point and is generally found around 20°S. At this time, strongest oxygen depletion is known to occur around ~24–26°S, while less severe oxygen depletion is observed near the ABF (Chapman and Shannon, 1987; Boyer et al., 2000). At the time of sampling, large horizontal gradients in SST and salinity existed around ~19.8ºS, fanning out seaward (Fig. 3b, c), indicating that the ABF had developed at this latitude. The ODZ did not extend past the frontal zone, and north of the ABF, oxygen depletion occurred

only down to ~20 µmol L$^{-1}$ (between 150–500 mbss; Fig. 4f), which would likely inhibit anammox (Strous et al., 1997; Woebken et al. 2007; Kalvelage et al., 2011). Coincidingly, evidence for the presence of *Ca.* Scalindua spp., as indicated by 16S rRNA gene sequences, BHT-*x* and IPL ladderane concentrations, was completely lacking at stations sampled north of the ABF (stations 18 and 59). The absence of anammox biomarkers here, is thus concurrent with the latitude of the ABF (~19.8°S) and the most severe oxygen depleted waters (~26°S), known to occur at this time. During austral winter, the ABF is located

furthest north (~14–16°S) and the most severe oxygen depletion occurs between 16–20°S (Chapman and Shannon, 1987; Boyer et al., 2000). Considering the seasonal northward shift of the ABF and oxygen-depleted waters during austral winter, the occurrence of anammox bacteria and associated biomarkers would likely shift northwards too, at this time of year.

At station 117, located just south of ABF, [O$_2$] was below ~20 µmol L$^{-1}$ at ~50 mbss (down to ~3 µmol L$^{-1}$ at 85 mbss; Fig. 5c), yet evidence of anammox was sparse. Station 117 was the only BUS station where N was not limited (N deficit

< 0 µmol L$^{-1}$; Fig. 4h; Table S2), revealing that loss of bioavailable nitrogen by anammox and/or denitrification was absent or limited here. No BHT-*x* was detected, and only very low abundances of ladderane IPLs ($8.6 \times 10^4$ ru L$^{-1}$; Fig. 5b) and *Ca.* Scalindua spp. 16S rRNA gene sequences (0.001‰; Fig. 5g) were detected at 85 mbss. Possibly, the water column at station 117 had only recently become oxygen depleted. If so, the slow growth rate of anammox bacteria (*e.g.* Strous et al., 1999; Jetten et al., 2009) could explain why biomarker and 16S rRNA gene evidence for the presence of anammox bacteria was sparse at

the time of sampling.

Salinity, temperature or nutrient (NO$_2^-$ and NH$_4^+$) concentrations were not seen to influence biomarker distributions in the BUS: i.e. no correlation was observed between these physiochemical parameters and BHT-*x* or ladderane IPLs (Table 2). This agrees with earlier findings. *Ca.* Scalindua spp. have an optimal temperature range of 10–30 ℃ (Awata et al., 2012; 2013), well within the temperature range found in the BUS. Furthermore, changes in salinity have not been found to affect

abundance of *Ca.* Scalindua spp. (Awata et al., 2012; 2013), and *Ca.* Scalindua spp. is known to have an extremely low affinity for NO$_2^-$ and NH$_4^+$ (Awata et al., 2013). In our study, only [O$_2$] returned a weak but significant negative correlation with BHT-*x* (r(60) = -0.33, $\rho$ = 0.01) and ladderane IPLs (r(60) = -0.29, $\rho$ = 0.02).

**Table 2.** Pearson's correlation matrix between BHT-*x*, oxygen (O$_2$), ammonium (NH$_4^+$), nitrite (NO$_2^-$), ladderane IPLs (IPLs), nitrogen
deficiency (N def), temperature (Temp), salinity (Sal) and BHT-*x*. *r*-values indicate Pearson's correlation coefficient. *p*-values indicate the significance level (2-tailed) with bold numbers indicating that correlation is significant at the 0.05 significance level.

| | | **BHT-*x*** | **IPLs** | **BHT-*x* ratio** | **NO$_2^-$** | **NH$_4^+$** | **N def.** | **Temp.** | **Sal.** | **O$_2$** |
|---|---|---|---|---|---|---|---|---|---|---|
| **BHT-*x*** | *r*-value | - | **0.65** | **0.59** | 0.12 | 0.17 | **0.53** | 0.02 | -0.01 | -0.33 |
| | $\rho$-value | | <0.001 | <0.001 | 0.36 | 0.19 | <0.001 | 0.88 | 0.92 | 0.01 |
| **IPLs** | *r*-value | **0.65** | - | **0.52** | -0.02 | -0.03 | **0.53** | 0.09 | 0.08 | **-0.29** |
| | $\rho$-value | <0.001 | | <0.001 | 0.88 | 0.83 | <0.001 | 0.51 | 0.57 | 0.02 |
| **BHT-*x* ratio** | *r*-value | **0.59** | **0.52** | - | -0.04 | 0.13 | **0.63** | -0.04 | -0.06 | **-0.35** |
| | $\rho$-value | <0.001 | <0.001 | | 0.74 | 0.31 | <0.001 | 0.76 | 0.66 | 0.01 |
| **NO$_2^-$** | *r*-value | 0.12 | -0.02 | -0.04 | - | **0.62** | 0.07 | 0.23 | **0.3** | **-0.3** |
| | $\rho$-value | 0.36 | 0.88 | 0.74 | | <0.001 | 0.61 | 0.08 | 0.02 | 0.02 |
| **NH$_4^+$** | *r*-value | 0.17 | -0.03 | 0.13 | **0.62** | - | **0.49** | 0.17 | 0.17 | **-0.32** |
| | $\rho$-value | 0.19 | 0.83 | 0.31 | <0.001 | | <0.001 | 0.20 | 0.19 | 0.01 |

| | | | | | | | | | | |
|---|---|---|---|---|---|---|---|---|---|---|
| **N def.** | *r*-value | **0.53** | **0.53** | **0.63** | 0.07 | **0.49** | - | 0.04 | -0.01 | **-0.37** |
| | *ρ*-value | <0.001 | <0.001 | <0.001 | 0.61 | <0.001 | | 0.76 | 0.95 | <0.005 |
| **Temp.** | *r*-value | 0.02 | 0.09 | -0.04 | 0.23 | 0.17 | 0.04 | - | **0.9** | -0.18 |
| | *ρ*-value | 0.88 | 0.512 | 0.759 | 0.076 | 0.197 | 0.757 | | <0.001 | 0.159 |
| **Sal.** | *r*-value | -0.01 | 0.08 | -0.06 | **0.3** | 0.17 | -0.01 | **0.9** | - | **-0.29** |
| | *ρ*-value | 0.92 | 0.57 | 0.66 | 0.03 | 0.19 | 0.95 | <0.001 | | 0.02 |
| **O₂** | *r*-value | **-0.33** | **-0.29** | **-0.35** | **-0.3** | **-0.32** | **-0.37** | -0.18 | **-0.29** | - |
| | *ρ*-value | 0.01 | 0.02 | 0.01 | 0.02 | 0.01 | <0.005 | 0.17 | 0.02 | |

### 4.1.3 Lateral transport of anammox biomarkers to oxygenated offshore waters

In the more oxygenated offshore waters (up to ~180 µmol L$^{-1}$), BHT-*x* was observed at stations 1, 2, 8 and 9, whereas ladderane IPLs were not detected and the relative abundance of the *Ca.* Scalindua spp. 16S rRNA gene was extremely low (Fig. 5). Potentially, ladderane IPLs (and hence, fresh anammox bacterial cells) were present, but simply below the detection limit of our method. However, various studies have reported a high sensitivity of IPLs when analysed using HPLC-ESI-MS. Especially IPLs with a PC headgroup were found to have a high response factor, likely due to the charged quaternary amine moiety on the PC headgroup (Sturt et al. 2003; van Mooy & Fredricks, 2010; Wörmer et al., 2015). In light of these results, it is unlikely that BHT-*x* was detected while the PC ladderane remained below the detection limit. Rather, it seems that a living anammox community was absent in offshore waters. Indeed, the offshore N deficit was limited (<4 µmol L$^{-1}$; Fig. 4d) and earlier reports (Kuypers et al., 2005) did not find anammox bacteria to be active in the BUS at an offshore station where bottom waters exceeded 20 µmol L$^{-1}$.

Yet, in apparent contradiction, high concentrations of ladderane FAs were detected at offshore station 2 at 125, 250 and 710 mbss, with peak concentrations at 250 mbss. In addition, BHT-*x* was observed at 250, 310 and 710 mbss, with the highest abundance found at the lowest depth (Fig. 7a). To determine the provenance of ladderane FAs observed at station 2, the NL$_5$ index was used (Table S5). The NL$_5$ index is correlated to the *in situ* growth temperature of anammox bacteria (Rattray et al. 2010). At station 2, NL$_5$ derived temperatures (21.0°C at 250 mbss and 15.6°C at 710 mbss) were substantially higher (i.e. ca. 10°C) than CTD-measured temperatures (11.2 and 5.5°C, respectively), indicating ladderane FAs were not synthesized *in situ*. In contrast, at shelf stations 6 and 140, NL$_5$ derived temperatures (7.4–15.0°C; between 30–85 mbss) were close to CTD temperature measurements (12.5–13.4°C; between 30–85 mbss), indicating *in situ* synthesis of ladderane FAs. This suggests that ladderane FAs observed offshore likely originated in the warmer shelf waters and were transported down-shelf.

Mollenhauer et al. (2007) showed that radiocarbon ages of lipid biomarkers in the BUS increased with distance from shore and water depth, as a consequence of lateral organic matter transport over the Namibian margin. In fact, most of the organic matter deposited offshore was found to derive from the shelf (Mollenhauer et al., 2007). In addition, Blumenberg et al., (2010) observed a decoupling of bio- and geohopanoids in BUS sediments, likely reflecting laterally-transported fossil organic matter. The degradation rate of ladderane FAs and BHT-*x* is slower than that of ladderane IPLs (i.e. ladderane FAs have been observed in sediments of 140 kyr BP; Jaeschke et al. 2009b, and BHT in sediments over 50 myr BP; Talbot et al., 2016, whereas ladderane IPLs are thought to reflect living or recently dead anammox cells; e.g. Jaeschke et al., 2009a). Subsequently, lateral offshore transport of organic matter in the nepheloid layer of the water column taking place in the BUS (Mollenhauer et al., 2007; Blumenberg et al., 2010), may have transported the more recalcitrant ladderane FAs and BHT-*x* from the ODZ on the shelf to offshore waters, whereas ladderane IPLs would not withstand this transport. Affirmatively, there is an absence of evidence for living anammox bacteria (e.g. 16S rRNA gene sequences and IPL ladderanes; Fig. 7a) at 125 and 250 mbss at station 2, further strongly suggesting an allochthonous origin of ladderane FAs and BHT-*x*.

### 4.1.4 Seasonality in anammox biomarker distributions

The Lüderitz upwelling cell has been identified as one of the most intense upwelling regions in the BUS. In austral winter, the water column near the cell is relatively oxygenated, due to the upwelling of oxygen-rich South Atlantic Central Water (Bailey et al., 1991). However, low-oxygen conditions and even anoxia prevail during austral summer due to the respiration of sinking organic matter supplied by phytoplankton blooms (Bailey et al., 1991; Brüchert et al., 2006). Consequently, continental shelf waters between 24–26°S display large temporal variations in DO concentrations under the influence of the Lüderitz upwelling

cell. At the time of sampling, the Lüderitz upwelling cell was apparent at ~26°S, appearing as a water mass with a low SST, low salinities (Fig. 3b, c) and high chlorophyll α concentrations (Table S1). Here, the water column was sampled once in February (station 6) and once in March (station 140), to explore the occurrence and distribution of anammox lipid biomarkers and 16S rRNA gene sequences, as the ODZ developed on the continental shelf (sediment depth 100 mbss).

In February (Fig. 7b), the nutrient, oxygen and temperature profiles show a highly stratified water column. A strong
oxycline is present around ~20 mbss, with near anoxic conditions in the bottom waters (down to ~3 µmol L$^{-1}$). 16S rRNA amplicon sequences of *Ca.* Scalindua spp. and BHT-*x* were detected below 40 mbss and 50 mbss, respectively, with (relative) abundances increasing with depth. Ladderane FAs followed a similar distribution, increasing in concentration with water column depth. However, ladderane IPLs were not detected throughout the water column, which may indicate that anammox bacteria were not yet a dominant feature in the water column community. Possibly, BHT-*x* and ladderane FAs at this station
were laterally transported from more southern shelf sites (Mollenhauer et al., 2007; Blumenberg et al., 2010). The accumulation of ammonium in the bottom waters (Fig. 7b), corresponding to a very high N deficit of 38 µmol L$^{-1}$ (Fig. 4h; Table S2), would suggest that denitrification was more active than anammox (Richards et al., 1965).

In March (Fig. 7c), the same sampling location showed distinct differences in physiochemical properties. This is consistent with previously reported seasonality: lower temperatures and increased upwelling commence in austral autumn,
resulting in decreased SSTs (Monteiro et al., 2008; Louw et al., 2016). Indeed, the strong redoxcline observed in February was absent in March. SST in March was also ~1.5°C lower than observed in February, indicating water column mixing and weakened stratification. Likewise, the nutrient-rich sub-thermocline waters mixed with the surface waters, resulting in similar NO$_2^-$, NO$_3^-$, and NH$_4^+$ concentrations throughout the water column. Additionally, salinity was relatively high throughout the water column (35.2–36.2 psu), indicating the late summer (Feb–April) salinity maximum (S > 35.1 psu) had set in, which is
known to co-occur with the oxygen minimum (Monteiro et al., 2008). Conformingly, in March, surface waters (<10 mbss) were more oxygen-depleted than observed in February. *Ca.* Scalindua spp. 16S rRNA genes were detected at all sampled depths, including 10 mbss. Likewise, ladderane IPLs, BHT-*x* and ladderane FAs were also present throughout the water column at all sampled depths (35–85 mbss). The ladderane FA and BHT-*x* concentrations were slightly lower then observed in February at 85 mbss, which may indicate that particulate material sank to the sea-floor, was degraded, or was transported
elsewhere prior to the occurrence of an established anammox community in March.

Our findings suggest a strong temporal variability in the presence of anammox bacteria and their synthesized lipids at 26°S, corresponding to a large shift in hydrographic characteristics of the water column. In all likelihood, anammox bacteria only became an established community at the end of austral summer, once the oxygen minimum had set in.

### 4.2 Application and constraints on the use of BHT-*x* as a biomarker for *Ca.* Scalindua

In the BUS, sequences taxonomically assigned to *Ca.* Scalindua spp. were detected at 11 out of 13 stations (Fig. 5g, n). A phylogenetically closely related cluster of *Ca.* Scalindua OTUs could be identified (i.e. the BUS OTU cluster indicated in Fig. 8). The BUS OTU cluster displayed a large sequence identity to *Ca.* Scalindua sorokinii isolated from the Guaymas deep sea hydrothermal vents and the Black Sea suboxic waters (98%; Table S7) and *Ca.* Scalindua brodae (97%; Table S7). BHT-*x* was originally reported to be uniquely synthesized by marine anammox, using *Ca.* S. brodae enrichment cultures (Rush et al., 2014;

Schwartz-Narbonne et al., 2020). In accordance with these reports, BHT-*x* was observed at the same 11 stations where *Ca.* Scalindua 16S rRNA gene sequences were detected.

At shelf stations (stations 3–6, 10, 117, 140), the presence of BHT-*x* co-occurred with the detection of *Ca.* Scalindua spp. 16S rRNA gene sequences at all depths except station 5 at 30 mbss. A multivariate binomial regression was performed to determine if the relative abundance of BHT-*x* can be used to predict the likelihood of 16S rRNA *Ca.* Scalindua spp. sequence presence in the BUS. This test showed that the presence of BHT-*x* significantly predicts the presence of *Ca.* Scalindua spp. in 78.8% of all cases ($\rho$ <0.001), showing BHT-*x* is a suitable biomarker for *Ca.* Scalindua spp. even in complex upwelling regions such as the BUS. However, in the BUS, *Ca.* Scalindua spp. 16S rRNA gene sequences constituted only a small portion of the total bacterial pool (max. 2.7‰; Fig. 5g). Low abundance of marine anammox bacteria in comparison to other phylogenetic groups in marine ecosystems has been reported previously (Woebken et al., 2007) and is likely caused by slow cell division rates (Strous et al., 1999; Jetten et al., 2009). Even so, it cannot be excluded that well-known PCR biases might also have led to a low coverage of *Ca.* Scalindua spp. reads. Unequal amplification efficiency of PCR products could result in the preferential amplification of certain 16S rRNA genes, whilst others might be inhibited for amplification (e.g. Pinto & Raskin, 2012).

*Ca.* Scalindua spp. 16S rRNA gene sequences were also detected in offshore waters. Yet, co-occurrence with BHT-*x* was limited (only in four of the 19 offshore SPM samples) and the extremely low relative abundance of *Ca.* Scalindua spp. 16S rRNA gene sequences here (0–0.4‰; Fig. 5n) and BHT-*x* concentrations (factor 10 to 100 lower than at shelf stations) make it unlikely that anammox bacteria formed an active community. Rather, lateral organic matter transport, discussed in section 4.1.3, seems to contribute to the BHT-*x* concentrations observed offshore. Considerations must thus be taken when interpreting low abundances of BHT-*x*, as these may inaccurately suggest the presence of living *Ca.* Scalindua.

## 4.3 Application and constraints on the use of BHT-*x* ratio as a biomarker for low oxygen conditions

In addition to being a useful biomarker for *Ca.* Scalindua, BHT-*x* has been applied as a proxy for low oxygen concentration in marine systems. Saénz et al (2011) proposed the ratio of BHT-*x* over total BHT as a proxy for suboxic-anoxic conditions (defined as [$O_2$] <5 µmol kg$^{-1}$), since BHT-*x* was only found in low-oxygen settings, whereas BHT is ubiquitously synthesized by mostly aerobic bacteria. The discovery that BHT-*x* is, so far known, uniquely synthesized by marine anammox (Rush et al., 2014; Schwarz-Narbonne et al., 2019), provided further evidence for this application of the BHT-*x* ratio, as it reflects the contribution of BHT-*x* synthesised by the anaerobic '*Ca.* Scalindua spp.' to the total BHT pool.

At BUS shelf stations, when [$O_2$] was >50 µmol L$^{-1}$, the BHT-*x* ratio remained below 0.04, in all but one case (station 5 at 30 mbss). However, at five offshore sites where [$O_2$] was >50 µmol L$^{-1}$ (up to ~180 µmol L$^{-1}$), BHT-*x* ratios >0.04 were observed. Likely, transported BHT-*x* derived from the ODZ on the shelf (see discussion section 4.1.3) and the markedly low BHT concentrations (Table S3), contributed to the relatively high BHT-*x* ratio signal observed here. When considering both offshore and on-shelf sites, when [$O_2$] was >50 µmol L$^{-1}$, the BHT-*x* ratio remained below 0.2, in all but one case (station 5 at 30 mbss). In addition, a ratio of ≥0.2 corresponded in all cases (except one) with the presence of the *Ca.* Scalindua spp. 16S rRNA gene, which was not the case for ratios >0.04. In the water columns of the Arabian Sea, Peru Margin and Cariaco Basin, Saénz et al. (2011) found that when [$O_2$] was >50 µmol kg$^{-1}$ (~50 µmol L$^{-1}$), the BHT-*x* ratio (i.e. BHT-II ratio) remained below 0.03 (with one exception). In Matys et al., (2017), highest BHT-*x* ratios (>0.2; i.e. BHT-II ratios) were detected in waters of the Humbold current system with [$O_2$] <3 µmol kg$^{-1}$. Yet, elevated ratios, thought to originate from transported organic matter, were also observed in oxygenated waters (> 200 µmol kg$^{-1}$) below the OMZ. Combining our results (BUS oxygen concentrations converted to µmol kg$^{-1}$; Fig. 9) with the findings of Sáenz et al., (2011) and Matys et al., (2017), show that that when [$O_2$] is <50 µmol kg$^{-1}$, the BHT-*x* ratio (i.e. BHT-II ratio) is ≥0.2 (except in 1 sample from the Cariaco Basin).

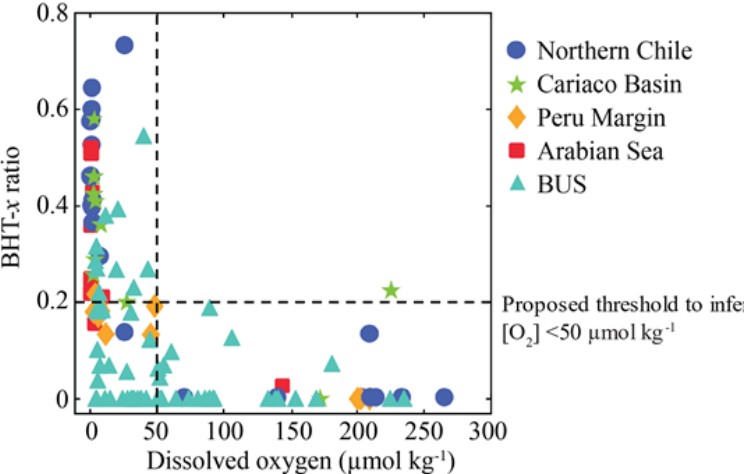

**Figure 9.** Relationship between dissolved oxygen concentration and the BHT-*x* ratio in suspended particulate matter (SPM) collected from the water columns of Northern Chile (Matys et al., 2017), the Cariaco Basin, Peru Margin and Arabian Sea (Sáenz et al., 2011) and the Benguela upwelling system (BUS; this study). Figure adapted from Sáenz et al., (2011) and Matys et al., (2017).

However, different extraction and/or analytical techniques may impact the BHT-*x* ratio. The BHT-isomer ratio derived from an acetylated culture analysed by Peiseler and Rohmer (1992) using HPLC (0.1), was different than that measured by Schwartz-Narbonne et al. (2019) in an aliquot of the same non-acetylated culture using UHPLC (0.2). Furthermore, to date, it is unknown if- and how the relative ratio of BHT and BHT-*x* synthesis by *Ca.* Scalindua spp. is influenced by environmental conditions and phylogeny. As BHT-*x* ratios observed in the BUS surpassed the ratio observed in the *Ca.* Scalindua brodae enrichment (Fig. 6), *Ca.* Scalindua spp. detected in the BUS might have synthesized BHT-*x* in a higher fractional abundance than observed in the enrichment. Even so, our results (using modified Bligh & Dyer extractions and UHPLC-HESI-MS) are found to align well with those from other marine systems, as investigated by Sáenz et al. (2011; Arabian Sea, Peru Margin and Cariaco Basin; using Soxhlet extractions and UHPLC-APCI-MS analysis) and Matys et al., (2017; Humboldt current system; using modified Bligh & Dyer extractions and UHPLC-APCI-MS analysis).

Nonetheless, in order to apply this threshold (BHT-*x* ratio ≥0.2) to infer low oxygen conditions (<50 µmol kg$^{-1}$) in sedimentary records, this signal must be retained in the sediment (i.e. not become diluted by BHT settling from the oxic zone of the water column). Matys et al. (2017) found that BHT II isomer ratios (i.e. BHT-*x* ratios) observed in surface sediments of the Humboldt current system were comparable to those observed in the OMZ core of the overlying water. In accordance, Berndmeyer et al., (2013) showed that BHPs recorded in the sediment of the Gotland Deep mirrored those of the suboxic zone of the water column. Hence, it is likely that BHT-*x* ratios observed in the low-oxygen zone of the water column, are retained in the underlying sediments. Moreover, considering the large variety in marine settings (four different upwelling regions and one restricted anoxic basin) and in methodologies, a BHT-*x* ratio of ≥0.2 is thought to provide a robust threshold in sedimentary records to estimate past low-oxygen conditions (<50 µmol kg$^{-1}$) of the overlying water column, hereby accounting for potential allochthonous BHT-*x* material.

**5 Conclusion**

This study reveals a strong spatiotemporal variability in the presence of anammox bacteria (as reflected by their 16S rRNA gene sequences) and their membrane lipids in the Benguela Upwelling System (BUS), which corresponds to differences in hydrographic characteristics of the water column. By elucidating the distribution of BHT-*x* across a large oxygen gradient, and comparing it to distributions of ladderane IPLs, ladderane FAs and *Ca.* Scalindua spp. 16S rRNA gene sequences, we assessed the suitability of BHT-*x* as a lipid biomarker for *Ca.* Scalindua spp., as well as its ratio over total BHT as a proxy for low-oxygen water column conditions. On the continental shelf, BHT-*x* co-occurred with the detection of *Ca.* Scalindua spp. 16S rRNA genes in all but one cases, further highlighting its suitability as a lipid biomarker for marine anammox in the sedimentary

record of upwelling regions. Shifts in the anammox lipid biomarker distribution at the southernmost shelf station (~25°S), sampled 27 days apart, implied that anammox bacteria only became an established community in the shelf waters at the end of austral summer, when oxygen depletion was most severe. At the offshore stations, ladderane FAs and low concentrations of BHT-$x$ were also observed to accumulate in relatively oxygenated waters ($[O_2]$ up to ~180 µmol L$^{-1}$), while ladderane IPLs were constrained to the shelf stations. Calculating the temperature sensitive NL$_5$ index for ladderane FAs, indicated that offshore ladderane FAs were not synthesized *in situ* and likely originated from the shelf. This must be taken into consideration when using BHT-$x$ and ladderane FAs as lipid biomarkers for *in situ* water column anammox. Lastly, at shelf stations, when $[O_2]$ was >50 µmol L$^{-1}$, the BHT-$x$ ratio remained below 0.04, in all but one case. Yet, laterally transported BHT-$x$ resulted in high offshore BHT-$x$ ratio values (>0.04) in oxygenated waters. We therefore suggest to use a BHT-$x$ ratio threshold of ≥0.2 to infer low oxygen conditions (<50 µmol kg$^{-1}$) in sedimentary records of dynamic upwelling systems: when comparing BUS BHT-$x$ (i.e. BHT-II) ratios with those from other marine settings (four different upwelling regions and one restricted anoxic basin; Saénz et al., 2011; Matys et al., 2017), it was observed that when $[O_2]$ was >50 µmol kg$^{-1}$, the BHT-$x$ ratio remained below 0.2 (in all but one case).

*Data availability.* Unassembled sequences are submitted to NCBI under BioProject number PRJNA761075. Individual sequences of OTUs 1-12 are published in GenBank under accession numbers OK086296 – OK086307.

*Supplement.* The supplement related to this article will be made available online (doi to be delivered later).

*Acknowledgements.* This research is supported by the Soehngen Institute of Anaerobic Microbiology (SIAM) Gravitation Grant (024.002.002) of the Netherlands Ministry of Education, Culture and Science (OCW) and the Netherlands Organization for Scientific Research (NWO) to J.S.S.D. and L.V. We kindly thank the captain and crew of the R/V *Pelagia* and the co-chief scientist on board, Dr. Zeynep Erdem, for facilitating the collection of all sampled material. Olga Żygadłowska is thanked for helping with the on-board sample processing and her great enthusiasm in doing so. We further greatly appreciate the help of Dr. Nicole Bale, with her knowledge on Bligh & Dyer extractions. Marianne Baas is thanked for deploying the *in situ* pumps during the second expedition. In addition, we thank Karel Bakker and Jan van Ooijen for the onboard NUTS analyses. We are also grateful for the support Denise Dorhout and Monique Verweij have delivered in the lipid lab and Maartje Brouwer and Sanne Vreugdenhil in the molecular labs. Lastly, Tom Vaessen is thanked for taking the time to discuss statistics.

*Author contribution.* ZRvK wrote the manuscript; PK and DR were in charge of the research expeditions; ZRvK, DR and PK performed the sample collection; ZRvK performed the laboratory work and data analysis. LV and HJW contributed to the data analysis of the 16S rRNA gene sequences; ECH optimized UHPLC measurements. ECH and DR contributed to the lipid data analysis. DR, JSSD, LV and ZRvK designed and conceptualized the project; All co-authors provided critical feedback and helped shape the research, analysis and manuscript.

*Competing interests.*
The authors declare that they have no conflict of interest.

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
