# Peer review of "Bacteriohopanetetrol-x: constraining its application as a lipid biomarker for marine anammox using the water column oxygen gradient of the Benguela upwelling system"

_Biogeosciences, 2021_

## Author Comment (AC1)

We greatly appreciate the time and effort referee 1 has taken to review our manuscript and are grateful for the positive assessment. The provided constructive comments will significantly contribute to a further improved manuscript.  Below, we will we will respond (**in bold**) to the reviewers' comments (*in italic*), point by point.

*"The authors present a threshold for the ratio of BHT-x/(BHT-x + BHT), which is introduced to infer "deoxygenation". They use the ratio, which they observed in their data sets (">0.04") and correct these ratio to ">0.18" to also account for potential complications from allochthonous organic matter (in challenging settings like the BUS). I agree that the latter process is an important issue, but it should be better explained what the ratio is exactly suggested for (most likely sedimentary studies), whether it can be transferred to other, so far unstudied settings"….(**see point 2 below**)…."A further complicating point if trying to establish a fixed threshold is the differences in extraction techniques (Soxhlet, ultrasonication, Bligh & Dyer), derivatisation (acetylated or not) and analytics (APCI vs. ESI). Considering these complications the authors should consider giving a less strict number like the suggested ">0.18", because it infers a high robustness or restrict the use to the here studied BUS setting."*

**We agree with the reviewer that establishing a BHT-x ratio threshold to infer deoxygenation poses multiple challenges. We will aim to better highlight these challenges in the revised manuscript, and to elaborate further on the proposed threshold. The following points will be discussed in further detail:**

1) ***The value of the BHT-x threshold value.* One of the aims of our study was to establish a threshold that can be used to determine past water column deoxygenation in sedimentary records of upwelling regions. As the reviewer notes, we established a relatively high BHT-x ratio threshold of 0.18 to infer water column oxygen levels of <50 $\mu$mol L$^{-1}$ to account for allochthonous anammox products. Nonetheless, as laid out by the reviewer, different extraction and/or analytical techniques may result in a different ratio. For instance, the BHT-*x* ratio derived from an acetylated culture analysed by Peiseler and Rohmer (1992) using HPLC (0.1), was different than that measured by Schwartz Narbonne et al. (2019) in an aliquot of the same non-acetylated culture using UHPLC (0.2). Thus, in accordance with the reviewer's comment, we will further emphasize that caution must be applied when comparing the BHT-x ratio between different studies with different methodologies. We therefore agree that a BHT-x ratio threshold of 0.18 determined in this study may be too constrained. Thus, in the revised manuscript we will remove a decimal point (rounding 0.18 up to 0.2). We show that our findings align well with those from different marine systems as investigated by Sáenz et al. (2011; Arabian Sea, Peru Margin and Cariaco Basin) *and* Matys et al., (2017; Humboldt current system). Combining these datasets with ours (oxygen concentrations converted to $\mu$mol kg$^{-1}$; see figure 1) indicates that when [O$_2$] is <50 $\mu$mol kg$^{-1}$, the BHT-x ratio (i.e. BHT-II ratio) is $\geq$ 0.2 (except in 1 sample from the Cariaco Basin). Considering the large variety in marine settings (four different upwelling regions and one restricted anoxic basin) and in methodologies (Soxhlet versus modified Bligh & Dyer; UHPLC-APCI-MS versus UHPLC-HESI-MS), a BHT-x ratio of 0.2 is likely to provide a robust threshold to estimate low-oxygen conditions (<50 $\mu$mol kg$^{-1}$) from sedimentary records. We will include this new figure in the revised manuscript, which shows the relationship between oxygen concentrations and the BHT-x ratio from all previous BHT-x water column studies.**

[Figure]

**Figure 1 (proposed new figure):** Relationship between dissolved oxygen concentration and the BHT-x ratio in suspended particulate matter (SPM) collected from the water columns of Northern Chile (Matys et al., 2017), the Cariaco Basin, Peru Margin and Arabian Sea (Sáenz et al., 2011) and the Benguela upwelling system (BUS; this study).

*.....”what it exactly tells us? Temporal or stable deoxygenation, deoxygenation in bottom waters or larger water bodies? Must the deoxygenation be occurring at a water body from which transport into the sediment is possible (sedimentary OM is not necessarily an integrated signal of SPM from all water depths). Further, I also ask the authors to cite and discuss a paper, which addressed the distribution of a BHT isomer (“BHT II”), which is tentatively the same as in Sáenz et al. (2011) and the study here, in a marine oxic-suboxic-anoxic water column and underlying sediments (Baltic Sea Gotland Deep; Berndmeyer et al., 2013). This papers shows that sedimentary OM only partly records water column SPM signals.”......*

2) ***Application of the BHT-x ratio.*** **As the reviewer points out, an integrated water column signal is not always recorded in the sedimentary record. We agree that the Berndmeyer et al. (2013) study valuably contributes to this discussion. Berndmeyer et al., (2013) show that bacteriohopanepolyols (BHPs) recorded in the sediment of the Gotland Deep are not an exact integrated signal of the entire water column, but instead more mirror the distinctive BHP distribution of the suboxic zone. In accordance, Matys et al. (2017) found that the BHT II isomer ratio (i.e. BHT-x ratio) values observed in surface sediments of the Humboldt current system are comparable to those observed in the OMZ core of the overlying water. This suggests that though BHPs found in sedimentary records might not be an integrated signal of the entire water column, it appears that the BHT-x and BHT-x ratio signal observed in the suboxic zones of the water column is preserved in the sediment. Though we have not analysed the BHT-x ratio in the underlying sediments of the BUS, based on the findings of Berndmeyer et al., (2013) and Matys et al., (2017), it is likely that the BHT-x ratio observed in the oxygen-deficient interval of the BUS water column is retained in the sediments. We aim to further discuss these facets in the discussion of the revised manuscript.**

*"The authors present a large and complicate multidisciplinary data set. Such a paper requires the best possible way of presentation. In general, the Figures are of high quality, but the map showing the sample locations is too small (and it does therefore not cover all information). Figure 2a should therefore be either enlarged or, better, presented as single Figure. Furthermore, station numbers should be better located in the Fig. at the respective symbols (and each stations should be labeled). It would then also be possible and helpful to add the profiles shown in Figure 7."*

**In accordance with reviewer #2's suggestion, we will aim to clarify the station map of figure 2a further by: i) separating figure 2a from the other figures (b, c and d), ii) enlarging the figure and iii) locating the station numbers nearer to their respective station location, as indicated with the dots. All station numbers for the CTD and nutrient measurements are provided in the supplementary material (with coordinates). The profiles of figure 7 will be presented earlier in the manuscript (after figure 2), and figure numbers will be adjusted accordingly.**

**"***At least at two places in the manuscript station numbers appear to be incorrect. At line 383 station "55" is mentioned, which is not in the Figures (potentially the authors refer to station 59?). At line 453 they refer to stations 8 and 55. It appears that both numbers are wrong here. The first is tentatively 18 and the second, again, 59. Station numbers given in the text must therefore be carefully checked!"*

**The reviewer is correct and we thank the referee for spotting these errors. We will correct this, and carefully check all station numbers provided in the text.**

*The authors use data from a natural setting and compare them with biomarker data from the laboratory. This is good and state of the art, but over interpretation of the lab data should be avoided. This holds also because only relatives of the organisms in the BUS water columns were available for lab studies and it remains unclear how valid these values are for the BUS (and other natural settings). For instance, using the BHT-x ratio from lab cultures would argue for (partly even more) than 100 % of bacterial hopanoid producers to be represented by anammox bacteria. This is unlikely and also far from the 16S rRNA data presented (less than 5 %). The authors discuss this discrepancy, but they should check whether not some of their statements need to be toned down. This refers also to the use of the temperature sensitive "NL5" ratios. The calculation is interesting and supports the conclusion of transported ladderane fatty acids, but decimal numbers for the temperature calculations appear to exact.*

**We agree that the discrepancy between BHT-x ratios found in the anammox biomass enrichment culture (cultivated in the laboratory) and the values observed in the BUS warrant further discussion. We will highlight potential discrepancies in the paragraph starting at line 567, by discussing that, to date, it is unknown if -and how BHT and BHT-x synthesis by *Ca.* Scalindua spp. is influenced by 1) environmental conditions and 2) species diversity. Concerning the $NL_5$ derived temperatures, we agree that the reported temperature values do not reflect the accuracy of the proxy. Thus, decimal numbers will be removed, as suggested by the reviewer.**

*I did not check all references, but there appears to be a discrepancy between references in the text and the reference list (e.g. Hopmans et al 2021 was not cited and Berndmeyer et al 2013 is in the list, but not in the text).*

**The Berndmeyer et al., 2014 study (2013 was not in the reference list) was incorrectly included in the reference list. We thank the reviewer for pointing this out and we will carefully check that only in text references are listed. The Hopmans et al., 2021 study is cited in the text in section '2.4.2 BHP and IPL analyses'.**

*Specific comments*

*Line 13: Modify for consistency to "(IPLs)"* **intact polar (IPL) ladderane lipids will be amended to ladderane intact polar lipids (IPLs).**

*Line 24: Change to "ratios"* **Amended.**

*Line 25: Introduce "NL5" here or rewrite.* **Amended.**

*Line 45: Delete part of the sentence from ", hereby…"* **Amended.**

*Line 56: Better deceased instead of "dead"?* **We feel that 'dead' is the more appropriate term for bacteria, and therefore propose to keep this term in the sentence.**

*Line 62: Is BHT-x really "rare"? In marine sediments with relatively high organic matter I would suppose not (e.g. in the Black Sea, the Cariaco Trench, the Baltic Sea this compound is abundantly reported).* **Agreed, the word 'rare' is removed from the sentence.**

*Line 63: I am not convinced that the current knowledge on the appearance of the BHT isomer allows describing it as "uniquely sourced by anammox". There is a convincing accord between anammox bacteria, their niches and BHT-x occurrences, but it does not exclude other sources. The authors may rethink the use of a less strict term here and elsewhere.* **Amended to "So far reported to be uniquely synthesized by marine anammox bacteria"**

*Line 81: here and elsewhere change to "Brüchert"* **Amended.**

*Line 135ff: Here the liters filtered should be added.* **Amended. The range in liters filtered will be added.**

*Line 159: The paper is not referenced in the list.* **The paper the citation refers to is number 40 in the reference list (Redfield et al., 1963), indeed, the year cited in the text was incorrect (1960 instead of 1963). This is now amended.**

*Line 193: "Hopmans et al 2021" is not in the reference list. I did not went through all references, but there appear to be inconsistencies. For instance, Berndmeyer et al 2013 is in the list, but not cited in the paper. This must be carefully checked and corrected!* **The Berndmeyer et al., 2014 study was indeed included in the reference list (but not the 2013 study) without any in-text reference. We thank the reviewer for pointing this out and will carefully check all references. The Hopmans et al., 2021 study is cited in the text in section '2.4.2 BHP and IPL analyses'.**

*Line 236 formula: For consistency write the denominator in brackets.* **Amended.**

*Line 243 and 245: Check symbol at "kit" and "Qiagen"* **Amended.**

*Line 289: Introduce "ABF" here.* **Amended.**

*Figure 3: Colors for station 8 and 9 are hard to distinguish. It is generally complicate to locate station-specific data in the biomarker plots. Why not using smaller symbol sizes, but also using different*

*symbols? What does "NB" in the legend means?* **Symbol size will be decreased and different colors will be used for station 8 and 9, to ensure they are distinguishable. NB is an abbreviation for 'Nota bene', i.e. Latin for 'note well'.**

*Line 316: Modify to "…near St. 117 or…"* **In this sentence, our intent was to indicate stations near the ABF (station 117) and north of the ABF (stations 18 and 59). We have amended this sentence to enhance clarity.**

*Line 321: Modify to "85 mbss"* **Amended.**

*Line 346: Modify to "were found in the BUS".* **Not all ladderane IPLs that were present in the anammox enrichment cultures were found in the BUS SPM samples. For clarity, we will amend this paragraph to read: "All the ladderane IPLs reported for the Ca. Scalindua brodae enrichment culture (Table S4) and those previously reported for Ca. Scalindua spp. (Rattray et al., 2008) were evaluated in the BUS SPM samples. However, at the time of sampling, only the PC and PG ladderanes (Fig. 1c) were detected in the BUS water column. Furthermore, these ladderane IPLs were found in SPM from a limited number of shelf stations located …"**

*Figure 5: Please give always the same x-axis for IPL-ladderanes (always 0 to 6 ru L-1). Also, why are numbers in Figure 3b so much higher (" x 10^5").* **We thank the reviewer for pointing this out, as indeed the figure is missing the factor by which the axis value should be multiplied. We will also provide the same scales for the ladderane IPL axis.**

*Line 473ff: Comment: BHT-x concentrations were also 10 less in the offshore samples. IPL ladderanes were not detected. However, is the sensitivity of both methods similar?*

**A study by Wörmer et al. (2015) provides a detailed overview of lipid biomarker analysis using HPLC/ESI-MS. Their results show a drastically expanded analytical window and sensitivity for IPLs when using reversed phase HPLC/ESI-MS, which we also applied here. In accordance, both Sturt et al. (2003) van Mooy & Fredricks (2010) report a high sensitivity for intact polar lipids (IPLs) using HPLC/ESI-MS. Though these latter two studies did not include analysis of BHPs, it is likely that the PC ladderane observed in our study has in fact a higher sensitivity than BHT (when analyzed using HPLC/ESI-MS), as the PC ladderane has a charged quaternary amine moiety, and therefore does not need to be ionized. The relative response factor of IPLs with a PC headgroup, in comparison to betaine lipids and glycolipids, was therefore observed to be relatively high (van Mooy & Fredricks, 2010). In addition, Wörmer et al., (2015) observed that IPLs with a PC headgroup had the highest response factor (and lowest ion suppression) in comparison to other IPLs. Nonetheless, it could be that ladderane IPLs at offshore stations were simply below the detection limit of our method. We aim to discuss all of the beforementioned points in further detail in the revised manuscript.**

*Line 485ff: Two publications should be added to this discussion, which reported on BHT-II in Benguela sediment (Watson, 2002) and on the problems of allochthonous organic matter in the same region (Blumenberg et al., 2010; geohopanoids including a "BHT (isomer 2", which is tentatively and and in analogy with the "BHT II" BHP in Sáenz et al. (2011) the BHT-x in this manuscript).* **Agreed. These publications will be included in the discussion.**

*Line 512ff: Sentence sounds odd and needs rewriting.* **Entire section from line 506 to 513 amended to:**

"**In March (Fig. 5c), the same sampling location showed distinct differences in physiochemical properties. This is consistent with previously reported seasonality: lower temperatures and increased upwelling commence in austral autumn, resulting in decreased SSTs (Monteiro et al.,**

2008; Louw et al., 2016).  Indeed, the strong redoxcline observed in February was absent in March. SST in March was also ~1.5°C lower than observed in February, indicating water column mixing and weakened stratification. Likewise, the nutrient-rich sub-thermocline waters mixed with the surface waters, resulting in similar $NO_2^-$, $NO_3^-$, and $NH_4^+$ concentrations throughout the water column (Fig. x). Additionally, salinity was relatively high throughout the water column (35.2–36.2 psu), indicating the late summer (Feb–April) salinity maximum (S > 35.1 psu) had set in, which is known to co-occur with the oxygen minimum (Monteiro et al., 2008). Indeed, in March, surface waters (<10 mbss) were more oxygen-depleted than observed in February.”

*Figure 7: Not sure, but there appears to be a discrepancy between the concentrations compared with Figure 3 (IPL ladderanes maximize in Fig. 3 at 2,5 x 10^5 and in Fig. 7 at 25 x 10^3). The authors should check that.* **The scale multiplication factor in figure 7 for the ladderane IPLs is corrected to 2.5x10^5 (i.e. 25x10^4), as the factor was indeed incorrect.**

*Line 564: An example, where a less exact threshold could be introduced. E.g. “…St. 5 at 30 mbss, and 0.2 may thus act as a safer threshold…”* **Amended.**

*Line 581: I don't think that the BHT-x ratio is correctly described as a marker for “anoxia”, but rather for anammox bacteria and its respective niches.* **Agreed, this will be rephrased**

*Line 584: Better modify to “…and indicate that anammox…”* **Amended.**

*Line 587: Better modify to “….the temperature sensitive NL5 index…”* **Amended.**

*Line 591: According to above, I recommend suggesting “0.2” instead of “0.18” here.* **Amended. See our comments above.**

*References: See general comment above and delete numbers for references.*

*Line 770: Requires splitting into two references.* **Amended.**

**References:**

1. **Berndmeyer, C., Thiel, V., Schmale, O. and Blumenberg, M.: Biomarkers for aerobic methanotrophy in the water column of the stratified Gotland Deep (Baltic Sea), Org. Geochem., 55, 103-111, doi: https://doi.org/10.1016/j.orggeochem.2012.11.010, 2013.**
2. **Louw, D. C., van der Plas, A. K., Mohrholz, V., Wasmund, N., Junker, T. and Eggert, A.: Seasonal and interannual phytoplankton dynamics and forcing mechanisms in the Northern Benguela upwelling system, J. Mar. Sy., 157, 124–134, doi:10.1016/j.jmarsys.2016.01.009, 2016.**
3. **Matys, E.D., Sepúlveda, J., Pantoja, S., Lange, C.B., Caniupán, M., Lamy, F. and Summons, R.E.: Bacteriohopanepolyols along redox gradients in the Humboldt Current System off northern Chile, Geobiology., (6), 844-857, doi: 10.1111/gbi.12250, 2017.**
4. **Monteiro, F. M., Pancost, R. D., Ridgwell, A. and Donnadieu, Y.: Nutrients as the dominant control on the spread of anoxia and euxinia across the Cenomanian-Turonian oceanic anoxic event (OAE2): Model-data comparison, Paleoceanography, 27(4), 1–17, doi:10.1029/2012PA002351, 2012.**
5. *Peiseler, B. and Rohmer, M.: Prokaryotic triterpenoids of the hopane series. Bacteriohopanetetrols of new side-chain configuration from Acetobacter species, J. Chem. Res., 298–299, 1992.*

6. Rattray, J. E., Van De Vossenberg, J., Hopmans, E. C., Kartal, B., Van Niftrik, L., Rijpstra, W. I. C., Strous, M., Jetten, M. S. M., Schouten, S. and Damsté, J. S. S.: Ladderane lipid distribution in four genera of anammox bacteria, Arch. Microbiol., 190(1), 51–66, doi:10.1007/s00203-008-0364-8, 2008.

7. Sáenz, J. P., Wakeham, S. G., Eglinton, T. I. and Summons, R. E.: New constraints on the provenance of hopanoids in the marine geologic record: Bacteriohopanepolyols in marine suboxic and anoxic environments, Org. Geochem., 42(11), 1351–1362, doi:10.1016/j.orggeochem.2011.08.016, 2011.

8. Schwartz-Narbonne, R., Schaeffer, P., Hopmans, E. C., Schenesse, M., Charlton, E. A., Jones, D. M., Sinninghe Damsté, J. S., Farhan Ul Haque, M., Jetten, M. S. M., Lengger, S. K., Murrell, J. C., Normand, P., Nuijten, G. H. L., Talbot, H. M. and Rush, D.: A unique bacteriohopanetetrol stereoisomer of marine 770 anammox, Org. Geochem., 143, doi:10.1016/j.orggeochem.2020.103994, 2020.

9. Sturt., H. F, Summons, R.E., Smith, K., Elvert, M., Hinrichs, K.: Intact polar membrane lipids in prokaryotes and sediments deciphered by high-performance liquid chromatography/electrospray ionization multistage mass spectrometry—new biomarkers for biogeochemistry and microbial ecology, Rapid Commun. Mass Spectrom., 18(6), 617–628, doi: https://doi.org/10.1002/rcm.1378, 2004.

10. Van Mooy, B.A.S and Fredricks, H.F.: Bacterial and eukaryotic intact polar lipids in the eastern subtropical South Pacific: Water-column distribution, planktonic sources, and fatty acid composition, Geochim., 74(22), 6499-6516, doi: https://doi.org/10.1016/j.gca.2010.08.026, 2010,

11. Wörmer, L., Lipp, J. S. and Hinrichs, K. U: Comprehensive analysis of microbial lipids in environmental samples through HPLC-MS protocols, In Hydrocarbon and lipid microbiology protocols, Springer, Berlin, Heidelber, 289-317, 2015.

---

## Author Comment (AC2)

We greatly appreciate the time and effort referee 2 has taken to review our manuscript and are grateful for the positive assessment. The provided constructive comments will significantly contribute to a further improved manuscript. Below, we will respond (**in bold**) to the reviewers' comments (*in italic*), point by point.

*Line 49. I would invert Figs. 1a and b, as ladderanes are presented first in the text.* **Amended.**

*Fig. 2 and Table 1, as well as materials and methods section. I do not understand why the sampling stations are not numbered consecutively. This should be explained somewhere.* **We thank the reviewer for pointing this out, the listing order of the various stations is indeed confusing. In Table 1, the reason for not listing the station numbers in a consecutive order was to highlight the division of 'shelf' and 'offshore' stations. In addition, within these two subdivisions, the stations are listed according to their sampling dates. We will highlight this in the caption of Table 1. Additionally, the odd numbering of the stations (e.g. jumping from station number 117 to 140) is due to the fact that during the second cruise (64PE450) stations were numbered according to activity (*e.g.* CTD sampling, multicoring etc.) rather than location. We will clarify this in the revised manuscript.**

*Line 171. "twice" instead of "thrice".* **Our intent with this phrasing was to clarify that after the first round of extraction, the supernatant was extracted three more times, where during the last two extractions the phosphate buffer was replaced with trichloroacetic acid. The phrasing might have been confusing, so we have rephrased this sentence to: "… re-extracted thrice (i.e. total of four extraction rounds), where during the last two extractions…"**

*Line 125. Please specify here how these standards were obtained (after having been isolated from sediments I imagine).* **The ladderane FAME standards were isolated from biomass of an anammox enrichment culture, grown in sequencing batch reactors, containing both *Ca.* Scalindua wagneri and *Ca.* Kuenenia stuttgartiensis (described in Kartal et al., 2006). We will add this information to the revised manuscript.**

*Line 322. Even 750 mbs for station 2.* **We have now included the specific bottom depths for St. 2 and 1 at which BHT-x was found.**

*Line 359. The point at 125 mbs is difficult to visualize.* **We will amend the figure (a.o. decrease symbol size) to clarify the visualization.**

*Line 383. Station 59 instead of 55.* **Amended.**

*Line 453. Station 18 and 59 instead?* **The reviewer is correct and is thanked for catching this error. Station number 8 is corrected to 18 and 55 to 59.**

*Lines 462-464. The seasonal effect should be better discussed here.* **We agree with the reviewer that a more detailed discussion about the seasonal shift of the Angola Benguela frontal zone and corresponding physicochemical changes in the water column would be appropriate. We propose to include the following text:**

**"The northern BUS is strongly influenced by the meeting of the warm, poleward flowing Angolan Current (AC) and the cold, equatorward flowing Benguela current (BC), which converge at the seasonally dynamic Angolan Benguela frontal zone. The balance between the intensities of the AC and BC determine the position of the front. At the end of austral summer (i.e. the timing of expeditions 64PE449 and 64PE450), the ABF reaches its most southern point and is generally**

found around 20°S. At this time, strongest oxygen depletion is known to occur around ~24-26°S, while less severe oxygen depletion is observed near the ABF. In contrast, during austral winter, the ABF is located furthest north (~14-16°S) and the most severe oxygen depletion occurs between 16-20°S (Chapman and Shannon, 1987; Boyer et al., 2000). Thus, the absence of anammox biomarkers north of the ABF during expeditions 64PE449 and 64PE450, is concurrent with the latitudes of the ABF (~19.8°S) and the most severe oxygen depleted waters (~26°S). Considering the seasonal northward shift of the ABF and the oxygen-depleted waterbodies, the occurrence of anammox bacteria and associated biomarkers will likely shift northwards too."

*Line 469. Affect abundance.* **Amended.**

*Lines 476-477. High concentrations in BHT-x were observed at 720 mbs at to a much lesser extent at 270m mbs, whereas the opposite was noted for ladderanes. This should be clearly specified.* **Amended to: "At St. 2, BHT-*x* was observed at 250, 310 and 710 mbss, with the highest abundance found at the lowest depth. Ladderane FAs were observed at 125, 250 and 710 mbss, with peak concentrations observed at 250 mbss."**

*Line 481. The persistence degree of ladderanes in the water column should be discussed here.* **The reviewer rightly points out that the degree of persistence of ladderane FAs in the water column is not properly discussed in this section. We propose to include the following text:**

**"The degradation rate of ladderane FAs is slower than that of ladderane IPLs (i.e. ladderane FAs have been observed in sediments of 140 kyr BP; Jaeschke et al. 2009b, whereas ladderane IPLs are thought to reflect living or recently dead anammox cells; *e.g.* Jaeschke et al., 2009a). Accordingly, ladderane FAs are likely not degraded immediately upon cell death and could be transported to other water bodies. Indeed, the offshore NL$_5$ derived temperatures suggest a higher *in situ* ambient temperature during ladderane FA synthesis than the CTD recorded temperatures. In contrast, NL$_5$ derived temperatures from shelf stations were similar to the CTD recorded temperatures, suggesting *in situ* synthesis of ladderane FAs on the shelf. This indicates that ladderane FAs observed offshore likely originated in the warmer shelf waters and were transported offshore."**

*Line 482. "likely indicating".* **Amended.**

*Line 493. I would define the Lüderitz upwelling cell here.* **We agree with the reviewer that the Lüderitz upwelling cell is not properly introduced here. We propose to amend line 493 to:**

**"The Lüderitz upwelling cell has been identified as one of the most intense upwelling regions in the BUS. In austral winter, the water column near the Lüderitz upwelling cell is relatively oxygenated, due to the upwelling of oxygen-rich South Atlantic Central Water (Bailey et al., 1991). However, low-oxygen conditions and even anoxia prevail during austral summer due to the respiration of sinking organic matter supplied by phytoplankton blooms (Bailey et al., 1991; Brüchert et al., 2006). Consequently, continental shelf waters between 24–26°S display large temporal variations in DO concentrations under the influence of the Lüderitz upwelling cell."**

*Lines 502-504. Here you should provide some hypotheses to explain why ladderane IPLs were not detected throughout the water column, whereas ladderane FA concentration increased with depth.*

*Where are ladderane FAs derived from? What about potential influence of lateral transport?* **Line 502-504 is amended to:**

**"Ladderane IPLs could have been present at St. 6 in abundances that were below the detection limit of our method. Alternatively, BHT-x and ladderane FAs at this station could have been laterally transported from more southern shelf sites (Mollenhauer et al., 2007), whereas ladderane IPLs which degrade quickly upon cell death (e.g. Jaeschke et al., 2009b) and would not withstand this transport."**

*Please check the salinity scale in Fig. 5c.* **We thank the reviewer for spotting this error in the number of decimal places, which should have been increased to two (giving 35.20, 35.25 and 35.30 as scale intervals). We will amend this in the revised manuscript.**

*Line 515. Similarly here, the relationship between ladderane IPLs and FAs should be better explained. Despite high abundance of ladderane IPLs, high abundance of ladderane FAs is not observed. This temporal offset should be discussed in more detail than just the sentence in lines 515-517.* **We will further highlight the potential explanations for the offset between ladderane IPL and ladderane FA concentrations.**

*Line 537. What do you mean by "well-known PCR biases"? This is unclear for the non-specialists.* **Amended to: "Unequal amplification efficiency of PCR products could result in the preferential amplification of certain 16S rRNA genes, whilst others might be inhibited for amplification (*e.g.* Pinto & Raskin, 2012). This could theoretically also have led to a low coverage of *Ca.* Scalindua spp. reads."**

*Fig. 7. The numbering in the caption and in the figure is not consistent.* **Amended.**

*Lines 563-566. This threshold should be tested in other sites, this could be mentioned.*

**We agree with the reviewer that comparing the BHT-x ratios observed in the BUS with BHT-x ratios observed at other sites would be a valuable contribution to the manuscript. We propose to include a new figure in the revised manuscript (see figure 1), which shows the relationship between oxygen concentrations and the BHT-x ratio in water columns from the Sáenz et al., (2011) and Matys et al., (2017) studies, as well as our own study. In order to compare our datasets, oxygen concentrations of our own study are converted to μmol kg$^{-1}$. Based on the data presented in this figure and the discussion in the rebuttal to referee 1, we propose rounding the threshold value up to 0.2. The combination of our own dataset with those of by Sáenz et al. (2011) and Matys et al., (2017), shows that when [O$_2$] is <50 μmol kg$^{-1}$, the BHT-x ratio is ≥ 0.2 (except 1 sample in the Cariaco Basin; see figure 1). Considering the large variety in marine settings (including four different upwelling regions and one restricted anoxic basin) and in methodologies (Soxhlet versus modified Bligh & Dyer; UHPLC-APCI-MS versus UHPLC-ESI-MS), we believe that a BHT-x ratio of 0.2 provides a robust threshold to estimate lox oxygen conditions (<50 μmol kg$^{-1}$) in sedimentary records of various marine settings, including upwelling regions.**

[Figure]

**Figure 1 (proposed new figure)**: Relationship between dissolved oxygen concentration and the BHT-x ratio in suspended particulate matter (SPM) collected from the water columns of Northern Chile (Matys et al., 2017), the Cariaco Basin, Peru Margin and Arabian Sea (Sáenz et al., 2011) and the Benguela upwelling system (BUS; this study).

**References:**

1. Bailey, G. W.: Organic carbon flux and development of oxygen deficiency on the modern Benguela continental shelf south of 22°S: Spatial and temporal variability, Geol. Soc. Spec. Publ., 58(58), 171–183, doi:10.1144/GSL.SP.1991.058.01.12, 1991.

2. Boyer, D., Cole, J. and Bartholomae, C.: Southwestern Africa: Northern Benguela Current region, Mar. Pollut. Bull., 41, 123–140, 2000.

3. Brüchert, V., Currie, B., Peard, K. R., Lass, U., Endler, R., Dübecke, A., Julies, E., Leipe, T. and Zitzmann, S.: Biogeochemical and Physical Control on Shelf Anoxia and Water Column Hydrogen Sulphide in the Benguela Coastal Upwelling System Off Namibia., 2006.

4. Chapman, P. and Shannon, L. V.: Seasonality in the oxygen minimum layers at the extremities of the Benguela system, South African J. Mar. Sci., 5(1), 85–94, doi:10.2989/025776187784522162, 1987.

5. Jaeschke, A., Rooks, C., Trimmer, M., Nicholls, J. C., Hopmans, E.C., Schouten, S. and Sinninghe Damsté, J.S.: Comparison of ladderane phospholipids and core lipids as indicators for anaerobic ammonium oxidation (anammox) in marine sediments, Geochim. Cosmochim. Acta, 73, 2077-2088, doi:10.1016/j.gca.2009.01.013, 2009a.

6. Jaeschke, A., Ziegler, M., Hopmans, E. C., Reichart, G. J., Lourens, L. J. and Schouten, S.: Molecular fossil evidence for anaerobic ammonium oxidation in the Arabian Sea over the last glacial cycle, Paleoceanography, 24(2), 1–11, doi:10.1029/2008PA001712, 2009b.

7. Kartal, B., Koleva, M., Arsov, R., van der Star, W., Jetten, M.S. and Strous, M. Adaptation of a freshwater anammox population to high salinity wastewater. J Biotechnol., 126(4), 546-53, doi: 10.1016/j.jbiotec.2006.05.012, 2006.

8. Matys, E.D., Sepúlveda, J., Pantoja, S., Lange, C.B., Caniupán, M., Lamy, F. and Summons, R.E. Bacteriohopanepolyols along redox gradients in the Humboldt Current System off northern Chile. Geobiology., (6), 844-857, doi: 10.1111/gbi.12250, 2017

9. Mollenhauer, G., Inthorn, M., Vogt, T., Zabel, M., Sinninghe Damsté, J. S. and Eglinton, T. I.: Aging of marine organic matter during cross-shelf lateral transport in the Benguela

upwelling system revealed by compound-specific radiocarbon dating, Geochemistry, Geophys. Geosystems, 8(9), doi:10.1029/2007GC001603, 2007.

10. Pinto, A.J. & Raskin, L.: PCR Biases Distort Bacterial and Archaeal Community Structure in Pyrosequencing Datasets, PLOS ONE, 7(8), e4309, doi:. 10.1371/journal.pone.0043093, 2012.

11. Sáenz, J. P., Wakeham, S. G., Eglinton, T. I. and Summons, R. E.: New constraints on the provenance of hopanoids in the marine geologic record: Bacteriohopanepolyols in marine suboxic and anoxic environments, Org. Geochem., 42(11), 1351–1362, doi:10.1016/j.orggeochem.2011.08.016, 2011.

---

## Author Response (AR1)

Dear editor,

Thank you for the evaluation of our manuscript. We have revised the manuscript according to the earlier proposed revisions, based on the suggestions and comments made by the referees. We greatly appreciate the time and effort both referees have taken to review our manuscript and are grateful for the positive assessment. The provided constructive comments have significantly contributed to a further improved manuscript. Below, we will indicate our response and the specific changes made in the revised manuscript (**in bold**) in relation to the reviewer's comments (*in italic*), point by point. In addition, the exact revisions can be viewed in the .pdf marked-up manuscript version, where the applied changes were tracked (in red).

**Referee #1**

*"The authors present a threshold for the ratio of BHT-x/(BHT-x + BHT), which is introduced to infer "deoxygenation". They use the ratio, which they observed in their data sets ("*>0.04*") and correct these ratio to "*>0.18*" to also account for potential complications from allochthonous organic matter (in challenging settings like the BUS). I agree that the latter process is an important issue, but it should be better explained what the ratio is exactly suggested for (most likely sedimentary studies), whether it can be transferred to other, so far unstudied settings"….(***see point 2 below***)…."A further complicating point if trying to establish a fixed threshold is the differences in extraction techniques (Soxhlet, ultrasonication, Bligh & Dyer), derivatisation (acetylated or not) and analytics (APCI vs. ESI). Considering these complications the authors should consider giving a less strict number like the suggested "*>0.18*", because it infers a high robustness or restrict the use to the here studied BUS setting."*

**We agree with the reviewer that establishing a BHT-x ratio threshold to infer deoxygenation poses multiple challenges. We have aimed to better highlight these challenges in the revised manuscript, and adjusted the text on the proposed threshold. One of the aims of our study was to establish a threshold that can be used to determine past water column deoxygenation in sedimentary records of upwelling regions. As the reviewer notes, we established a relatively high BHT-x ratio threshold of 0.18 to infer water column oxygen levels of <50 μmol L$^{-1}$ to account for allochthonous anammox products. Nonetheless, as laid out by the reviewer, different extraction and/or analytical techniques may result in a different ratio. For instance, the BHT-*x* ratio derived from an acetylated culture analysed by Peiseler and Rohmer (1992) using HPLC (0.1), was different than that measured by Schwartz Narbonne et al. (2019) in an aliquot of the same non-acetylated culture using UHPLC (0.2). Thus, in accordance with the reviewer's comment, we will further emphasize that caution must be applied when comparing the BHT-x ratio between different studies with different methodologies. We therefore agree that a BHT-x ratio threshold of 0.18 determined in this study may be too constrained. Thus, in the revised manuscript we have removed one significant number (e.g., rounding 0.18 up to 0.2). We show that our findings align well with those from different marine systems as investigated by Sáenz et al. (2011; Arabian Sea, Peru Margin and Cariaco Basin) *and* Matys et al., (2017; Humboldt current system). Combining these datasets with ours (oxygen concentrations converted to μmol kg$^{-1}$; see figure 1) indicates that when [O$_2$] is <50 μmol kg$^{-1}$, the BHT-x ratio (i.e. BHT-II ratio) is ≥ 0.2 (except in 1 sample from the Cariaco Basin). Considering the large variety in marine settings (four different upwelling regions and one restricted anoxic basin) and in methodologies (Soxhlet versus modified Bligh & Dyer; UHPLC-APCI-MS versus UHPLC-HESI-MS), a BHT-x ratio of 0.2 is likely to provide a robust threshold to estimate low-oxygen conditions (<50 μmol kg$^{-1}$) from sedimentary records. Accordingly, we have included a new figure in the revised manuscript, (i.e., Fig. 9), which shows the relationship between oxygen concentrations and the BHT-x ratio from all previous BHT-x water column studies. Discussion section 4.3. is completely amended to include this comment of the referee.**

[Figure]

**Figure 9:** Relationship between dissolved oxygen concentration and the BHT-*x* ratio in suspended particulate matter (SPM) collected from the water columns of Northern Chile (Matys et al., 2017), the Cariaco Basin, Peru Margin and Arabian Sea (Sáenz et al., 2011) and the Benguela upwelling system (BUS; this study). Figure adapted from Sáenz et al., (2011) and Matys et al., (2017).

50

*......"what it exactly tells us? Temporal or stable deoxygenation, deoxygenation in bottom waters or larger water bodies? Must the deoxygenation be occurring at a water body from which transport into the sediment is possible (sedimentary OM is not necessarily an integrated signal of SPM from all water depths). Further, I also ask the authors to cite and discuss a paper, which addressed the distribution of a BHT isomer ("BHT II"), which is*

55 *tentatively the same as in Sáenz et al. (2011) and the study here, in a marine oxic-suboxic-anoxic water column and underlying sediments (Baltic Sea Gotland Deep; Berndmeyer et al., 2013). This papers shows that sedimentary OM only partly records water column SPM signals."*

**With respect to the application of the BHT-x ratio, we agree with the reviewer that an integrated water**
60 **column signal is not always recorded in the sedimentary record. We also agree that the Berndmeyer et al. (2013) study valuably contributes to this discussion. Berndmeyer et al., (2013) show that bacteriohopanepolyols (BHPs) recorded in the sediment of the Gotland Deep are not an exact integrated signal of the entire water column, but instead mirror the distinctive BHP distribution of the suboxic zone. In accordance, Matys et al. (2017) found that the BHT II isomer ratio (i.e. BHT-x ratio) values observed in surface**
65 **sediments of the Humboldt current system are comparable to those observed in the OMZ core of the overlying water. This suggests that though BHPs found in sedimentary records might not be an integrated signal of the entire water column, it appears that the BHT-x and BHT-x ratio signal observed in the suboxic zones of the water column is preserved in the sediment. Though we have not analysed the BHT-x ratio in the underlying sediments of the BUS, based on the findings of Berndmeyer et al., (2013) and Matys et al., (2017),**
70 **it is likely that the BHT-x ratio observed in the oxygen-deficient interval of the BUS water column is retained in the sediments. We have further discussed these aspect in the discussion of the revised manuscript (i.e., section 4.3).**

*"The authors present a large and complicate multidisciplinary data set. Such a paper requires the best possible*
75 *way of presentation. In general, the Figures are of high quality, but the map showing the sample locations is too small (and it does therefore not cover all information). Figure 2a should therefore be either enlarged or, better, presented as single Figure. Furthermore, station numbers should be better located in the Fig. at the respective symbols (and each stations should be labeled). It would then also be possible and helpful to add the profiles shown in Figure 7."*

80 **In accordance with reviewer #2's suggestion, we have clarified the station map of figure 2a further by: i) separating figure 2a from the other figures (b, c and d) and ii) enlarging the figure. All station numbers for the**

**CTD and nutrient measurements are provided in the supplementary material (with coordinates). The profiles of figure 7 will be presented earlier in the manuscript (after figure 2, i.e. figure 3 in revised manuscript), and figure numbers have been adjusted accordingly.**

*"At least at two places in the manuscript station numbers appear to be incorrect. At line 383 station "55" is mentioned, which is not in the Figures (potentially the authors refer to station 59?). At line 453 they refer to stations 8 and 55. It appears that both numbers are wrong here. The first is tentatively 18 and the second, again, 59. Station numbers given in the text must therefore be carefully checked!"*

**The reviewer is correct and we thank the referee for spotting these errors. We have corrected this (line 383: "….ABF (St. 18 and 59)…."), and carefully checked all station numbers provided in the text.**

*The authors use data from a natural setting and compare them with biomarker data from the laboratory. This is good and state of the art, but over interpretation of the lab data should be avoided. This holds also because only relatives of the organisms in the BUS water columns were available for lab studies and it remains unclear how valid these values are for the BUS (and other natural settings). For instance, using the BHT-x ratio from lab cultures would argue for (partly even more) than 100 % of bacterial hopanoid producers to be represented by anammox bacteria. This is unlikely and also far from the 16S rRNA data presented (less than 5 %). The authors discuss this discrepancy, but they should check whether not some of their statements need to be toned down. This refers also to the use of the temperature sensitive "NL5" ratios. The calculation is interesting and supports the conclusion of transported ladderane fatty acids, but decimal numbers for the temperature calculations appear to exact.*

**We agree that the discrepancy between BHT-x ratios found in the anammox biomass enrichment culture (cultivated in the laboratory) and the values observed in the BUS warrant further discussion. We have highlighted potential discrepancies in section 4.3, by discussing that, to date, it is unknown if -and how BHT and BHT-x synthesis by *Ca.* Scalindua spp. is influenced by 1) environmental conditions and 2) species diversity. Concerning the $NL_5$ derived temperatures, we agree that the reported temperature values do not reflect the accuracy of the proxy. Thus, decimal numbers have been removed, as suggested by the reviewer. In addition, ladderane fatty acid concentrations have been converted from ng $L^{-1}$ to pg $L^{-1}$.**

*I did not check all references, but there appears to be a discrepancy between references in the text and the reference list (e.g. Hopmans et al 2021 was not cited and Berndmeyer et al 2013 is in the list, but not in the text).*

**The Berndmeyer et al., 2014 study (2013 was not in the reference list) was incorrectly included in the reference list. We thank the reviewer for pointing this out. We carefully checked all references, and made the appropriate corrections. The Hopmans et al., 2021 study is cited in the text in section '2.4.2 BHP and IPL analyses'.**

*Specific comments*

*Line 13: Modify for consistency to "(IPLs)"* **intact polar (IPL) ladderane lipids is amended to ladderane intact polar lipids (IPLs).**

*Line 24: Change to "ratios"* **Amended.**

*Line 25: Introduce "NL5" here or rewrite.* **Amended to: "…were undetected. The index of ladderane lipids with five cyclobutane rings ($NL_5$) correlates with *in situ* temperature. $NL_5$ derived…"**

*Line 45: Delete part of the sentence from ", hereby…"* **Amended to: "Climate models predict that OMZs will expand both spatially and temporally (Oschlies et al., 2018), hereby altering the biogeochemistry of the oceans. This will likely increase the potential of fixed N-loss processes, such as anammox, in marine systems (Breitburg et al. 2018)."**

*Line 56: Better deceased instead of "dead"?*

**We feel that 'dead' is the more appropriate term for bacteria, and therefore kept this term.**

*Line 62: Is BHT-x really "rare"? In marine sediments with relatively high organic matter I would suppose not (e.g. in the Black Sea, the Cariaco Trench, the Baltic Sea this compound is abundantly reported).* **Agreed, the word 'rare' is removed from the sentence.**

*Line 63: I am not convinced that the current knowledge on the appearance of the BHT isomer allows describing it as "uniquely sourced by anammox". There is a convincing accord between anammox bacteria, their niches and BHT-x occurrences, but it does not exclude other sources. The authors may rethink the use of a less strict term here and elsewhere.* **Amended to:**

**Section 4.2, line 526: "…BHT-*x* was originally reported to be uniquely synthesized…"**

**Section 4.3, line 553: "…that BHT-*x* is, so far known, uniquely synthesized…"**

*Line 81: here and elsewhere change to "Brüchert"* **Amended to:**

**Line 81: "…Verheye, 2005; Brüchert et al., 2006…"**

**Line 112: "…1999; Bruchert et al., 2006…"**

**Line 115: "…Verheye, 2005; Brüchert et al., 2006…"**

*Line 135ff: Here the liters filtered should be added.* **Amended to:  "Suspended particulate matter (SPM) for lipid analysis was collected using four McLane Large Volume Water Transfer System Sampler (WTS-LV) in situ pumps, which were deployed for 4h (~40-900 L filtered; McLane Laboratories Inc., Falmouth, MA, USA)."**

*Line 159: The paper is not referenced in the list.* **The paper the citation refers to is Redfield et al., (1963). Indeed, the year cited in the text was incorrect (1960 instead of 1963). This is now amended to: "….N:P (Redfield et al., 1963)…."**

*Line 193: "Hopmans et al 2021" is not in the reference list. I did not went through all references, but there appear to be inconsistencies. For instance, Berndmeyer et al 2013 is in the list, but not cited in the paper. This must be carefully checked and corrected!* **The Berndmeyer et al., 2014 study was indeed included in the reference list (but not the 2013 study) without any in-text reference. We thank the reviewer for pointing this out. The Hopmans et al., 2021 study is cited in the text in section '2.4.2 BHP and IPL analyses'. All references were carefully checked and corrected.**

*Line 236 formula: For consistency write the denominator in brackets.* **Amended. Denominators in equation 3 and 4 are now given in brackets.**

*Line 243 and 245: Check symbol at "kit" and "Qiagen"* **Amended to:**

**"…DNeasy Powersoil kit®. PCR…"**

**"…with the Qiagen® PCR reagents…"**

*Line 289: Introduce "ABF" here.* **Amended to: "…The location of the Angolan Benguela front (ABF) during the…"**

*Figure 3: Colors for station 8 and 9 are hard to distinguish. It is generally complicate to locate station-specific data in the biomarker plots. Why not using smaller symbol sizes, but also using different symbols? What does "NB" in the legend means?* **Figure now includes smaller symbol sizes (80% of original size) and colors for station 8 and 59 have been changed to yellow and orange respectively. NB is an abbreviation for 'Nota bene', i.e. Latin for 'note well'.**

*Line 316: Modify to "…near St. 117 or…"* **In this sentence, our intent was to indicate stations near the ABF (station 117) and north of the ABF (stations 18 and 59). To enhance clarity, the sentence is amended to: "No BHT-*x* was observed at stations located near the ABF (St. 117) nor north of the ABF (St. 18 and 59)."**

*Line 321: Modify to "85 mbss".* **Amended to: "….at 85 mbss (Fig. 6b)…."**

170 *Line 346: Modify to "were found in the BUS".* **Not all ladderane IPLs that were present in the anammox enrichment cultures were found in the BUS SPM samples. For clarity, we amended this paragraph to: "All the ladderane IPLs reported for the *Ca*. Scalindua brodae enrichment culture (Table S4) and those previously reported for Ca. Scalindua spp. (Rattray et al., 2008) were evaluated in the BUS SPM samples. However, at the time of sampling, only the PC and PG ladderanes (Fig. 1c) were detected in the BUS water column.**
175 **Furthermore, these ladderane IPLs were found in SPM from a limited number of shelf stations located …"**

*Figure 5: Please give always the same x-axis for IPL-ladderanes (always 0 to 6 ru L-1). Also, why are numbers in Figure 3b so much higher (" x 10^5").* **We thank the reviewer for pointing this out, as indeed the figure is missing the factor by which the axis value should be multiplied. We have amended the figure as follows: in figure 5a and b (in revised manuscript figure 7a and b), the axis for the ladderane IPLs are changed (i.e. the**
180 **scale is removed and axis only contain 'N.D.', indicating 'not detected', to clarify that ladderane IPL concentrations were below the detection limit at these stations). Furthermore, all symbol sizes are decreased (to 80% of original symbol size) and the outline of the symbols is removed. Lastly, 'IPL ladderanes' is changed to 'Ladderane IPLs'.**

*Line 473ff: Comment: BHT-x concentrations were also 10 less in the offshore samples. IPL ladderanes were not*
185 *detected. However, is the sensitivity of both methods similar?*

**A study by Wörmer et al. (2015) provides a detailed overview of lipid biomarker analysis using HPLC/ESI-MS. Their results show a drastically expanded analytical window and sensitivity for IPLs when using reversed phase HPLC/ESI-MS, which we also applied here. In accordance, both Sturt et al. (2003) van Mooy & Fredricks (2010) report a high sensitivity for intact polar lipids (IPLs) using HPLC/ESI-MS. Though these latter two**
190 **studies did not include analysis of BHPs, it is likely that the PC ladderane observed in our study has in fact a higher sensitivity than BHT (when analyzed using HPLC/ESI-MS), as the PC ladderane has a charged quaternary amine moiety, and therefore does not need to be ionized. The relative response factor of IPLs with a PC headgroup, in comparison to betaine lipids and glycolipids, was therefore observed to be relatively high (van Mooy & Fredricks, 2010). In addition, Wörmer et al., (2015) observed that IPLs with a PC headgroup**
195 **had the highest response factor (and lowest ion suppression) in comparison to other IPLs. Nonetheless, it could be that ladderane IPLs at offshore stations were simply below the detection limit of our method. We now discuss all of the beforementioned points in Discussion section 4.1.3.**

*Line 485ff: Two publications should be added to this discussion, which reported on BHT-II in Benguela sediment*
200 *(Watson, 2002) and on the problems of allochthonous organic matter in the same region (Blumenberg et al., 2010; geohopanoids including a "BHT (isomer 2", which is tentatively and and in analogy with the "BHT II" BHP in Sáenz et al. (2011) the BHT-x in this manuscript).* **Agreed. These publications are now included in the discussion:**

**Section 41.3, Line 485: "….et al., 2007). In addition, Blumenberg et al., (2010) observed a decoupling of bio-**
205 **and geohopanoids in BUS sediments, likely reflecting laterally-transported fossil organic matter. Thus, lateral…."**

*Line 512ff: Sentence sounds odd and needs rewriting.* **Entire section from line 506 to 513 amended to:**

**"In March (Fig. 7c), the same sampling location showed distinct differences in physiochemical properties. This is consistent with previously reported seasonality: lower temperatures and increased upwelling commence in**
210 **austral autumn, resulting in decreased SSTs (Monteiro et al., 2008; Louw et al., 2016). Indeed, the strong redoxcline observed in February was absent in March. SST in March was also ~1.5°C lower than observed in**

February, indicating water column mixing and weakened stratification. Likewise, the nutrient-rich sub-thermocline waters mixed with the surface waters, resulting in similar $NO_2^-$, $NO_3^-$, and $NH_4^+$ concentrations throughout the water column. Additionally, salinity was relatively high throughout the water column (35.2–36.2 psu), indicating the late summer (Feb–April) salinity maximum (S > 35.1 psu) had set in, which is known to co-occur with the oxygen minimum (Monteiro et al., 2008). Indeed, in March, surface waters (<10 mbss) were more oxygen-depleted than observed in February.”

*Figure 7: Not sure, but there appears to be a discrepancy between the concentrations compared with Figure 3 (IPL ladderanes maximize in Fig. 3 at 2,5 x 10^5 and in Fig. 7 at 25 x 10^3). The authors should check that.* **The scale multiplication factor in figure 7 (i.e. figure 4 in revised manuscript) for the ladderane IPLs is corrected to 25x10^4, as the factor was indeed incorrect.**

*Line 564: An example, where a less exact threshold could be introduced. E.g. “…St. 5 at 30 mbss, and 0.2 may thus act as a safer threshold…”* **Amended. BHT-x ratios of 0.18 are rounded to 0.2 throughout the text. See point one for exact text in this section.**

*Line 581: I don't think that the BHT-x ratio is correctly described as a marker for "anoxia", but rather for anammox bacteria and its respective niches.* **Agreed, this is now rephrased to: “….total BHT as a proxy for low-oxygen water column conditions. On the…”**

*Line 584: Better modify to “…and indicate that anammox…”* **Amended to:**

**“…upwelling regions. Shifts in the anammox lipid biomarker distribution at the southernmost shelf station (~25°S), sampled 27 days apart, implied that anammox bacteria only became an established community in the shelf waters at the end of austral summer, when oxygen depletion was most severe. At the…”**

*Line 587: Better modify to “….the temperature sensitive NL5 index…”* **Amended to:**

**“…shelf stations. Calculating the temperature sensitive $NL_5$ index for ladderane FAs, indicated that…”**

*Line 591: According to above, I recommend suggesting “0.2” instead of “0.18” here.* **Amended. See our comments above.**

*References: See general comment above and delete numbers for references.* **Amended.**

*Line 770: Requires splitting into two references.* **Amended.**

**Referee #2**

*Line 49. I would invert Figs. 1a and b, as ladderanes are presented first in the text.* **Amended.**

*Fig. 2 and Table 1, as well as materials and methods section. I do not understand why the sampling stations are not numbered consecutively. This should be explained somewhere.* **In Table 1, the reason for not listing the station numbers in a consecutive order was to highlight the division of ‘shelf' and ‘offshore' stations. In addition, within these two subdivisions, the stations are listed according to their sampling dates. We have now highlighted this in the caption of Table 1. Additionally, the odd numbering of the stations (e.g. jumping from station number 117 to 140) is due to the fact that during the second cruise (64PE450) stations were numbered according to activity (*e.g.* CTD sampling, multicoring etc.) rather than location. To allow for comparison of station numbers between this paper and potential future papers from other authors, we decided to keep the station numbering the way it was determined during the expedition. The caption of table 1 has been amended to:**

**Caption table 1: “…Stations are grouped according to their location on the continental shelf and listed based on their sampling date. Location is ‘shelf'…”**

*Line 171. "twice" instead of "thrice".* **Our intent with this phrasing was to clarify that after the first round of extraction, the supernatant was extracted three more times, where during the last two extractions the phosphate buffer was replaced with trichloroacetic acid. The phrasing might have been confusing, so we have rephrased this sentence to: "… re-extracted thrice (i.e. total of four extraction rounds), where during the last two extractions, the phosphate buffer was replaced with a trichloroacetic acid solution to enable optimal recovery of IPLs (Sturt et al., 2004). Phase separation…"**

*Line 125. Please specify here how these standards were obtained (after having been isolated from sediments I imagine).* **The ladderane FAME standards were isolated from biomass of an anammox enrichment culture, grown in sequencing batch reactors, containing both *Ca.* Scalindua wagneri and *Ca.* Kuenenia stuttgartiensis (described in Kartal et al., 2006). We have added this information to the revised manuscript as follows:**

**Line 125: "…2011). Ladderane FAME standards were isolated from an anammox enrichment culture grown in sequencing batch reactors, containing both *Ca.* Scalindua wagneri and *Ca.* Kuenenia stuttgartiensis (described in Kartal et al., 2006). The index…"**

*Line 322. Even 750 mbs for station 2.* **We have now included the specific bottom depths for St. 2 and 1 at which BHT-x was found as follows:**

**Line 322: "…mbss and in the bottom waters of station 1 (1500 mbss) and 2 (710 mbss). Here, the…"**

*Line 359. The point at 125 mbs is difficult to visualize.* **Symbol sizes in figure 3 (figure 5 in revised manuscript) are decreased to 80% of original symbol size.**

*Line 383. Station 59 instead of 55.* **Amended to 'St. 59'**

*Line 453. Station 18 and 59 instead?* **The reviewer is correct and is thanked for catching this error. Station number 8 is corrected to 18 and 55 to 59.**

*Lines 462-464. The seasonal effect should be better discussed here.* **We agree with the reviewer that a more detailed discussion about the seasonal shift of the Angola Benguela frontal zone and corresponding physicochemical changes in the water column would be appropriate. Section 4.1.2 is now amended to include the following text:**

**"At the end of austral summer (i.e. the timing of expeditions 64PE449 and 64PE450), the ABF reaches its most southern point and is generally found around 20°S. At this time, strongest oxygen depletion is known to occur around ~24-26°S, while less severe oxygen depletion is observed near the ABF. At the time of sampling……**

**……….(stations 18 and 59). The absence of anammox biomarkers here, is thus concurrent with the latitude of the ABF (~19.8°S) and the most severe oxygen depleted waters (~26°S), known to occur at this time. During austral winter, the ABF is located furthest north (~14-16°S) and the most severe oxygen depletion occurs between 16-20°S (Chapman and Shannon, 1987; Boyer et al., 2000). Considering the seasonal northward shift of the ABF and the oxygen-depleted waterbodies during austral winter, the occurrence of anammox bacteria and associated biomarkers would likely shift northwards too at this time of year."**

*Line 469. Affect abundance.* **Amended to: "…found to affect abundance of *Ca…..*"**

*Lines 476-477. High concentrations in BHT-x were observed at 720 mbs at to a much lesser extent at 270m mbs, whereas the opposite was noted for ladderanes. This should be clearly specified.* **Amended to:**

**Line 476: "Yet, in apparent contradiction, high concentrations of ladderane FAs were detected at offshore St. 2 at 125, 250 and 710 mbss, with peak concentrations at 250 mbss . In addition, BHT-*x* was observed at 250, 310 and 710 mbss, with the highest abundance found at the lowest depth (Fig. 7a). To determine.."**

*Line 481. The persistence degree of ladderanes in the water column should be discussed here.* **The reviewer rightly points out that the degree of persistence of ladderane FAs in the water column is not properly discussed in this section. This section has been amended to include the following text:**

**"The degradation rate of ladderane FAs and BHT-x is slower than that of ladderane IPLs (i.e. ladderane FAs have been observed in sediments of 140 kyr BP; Jaeschke et al. 2009b, and BHT in sediments over 50 myr BP; Talbot et al., 2016, whereas ladderane IPLs are thought to reflect living or recently dead anammox cells; e.g. Jaeschke et al., 2009a). Subsequently, lateral offshore transport of organic matter in the nepheloid layer of the water column taking place in the BUS (Mollenhauer et al., 2007; Blumenberg et al., 2010), may have transported the more recalcitrant ladderane FAs and BHT-x from the ODZ on the shelf to offshore waters, whereas ladderane IPLs would not withstand this transport."**

*Line 482. "likely indicating".* **Amended.**

*Line 493. I would define the Lüderitz upwelling cell here.* **We agree with the reviewer that the Lüderitz upwelling cell is not properly introduced here. We have amended line 493 to:**

**"The Lüderitz upwelling cell has been identified as one of the most intense upwelling regions in the BUS. In austral winter, the water column near the Lüderitz upwelling cell is relatively oxygenated, due to the upwelling of oxygen-rich South Atlantic Central Water (Bailey et al., 1991). However, low-oxygen conditions and even anoxia prevail during austral summer due to the respiration of sinking organic matter supplied by phytoplankton blooms (Bailey et al., 1991; Brüchert et al., 2006). Consequently, continental shelf waters between 24–26°S display large temporal variations in DO concentrations under the influence of the Lüderitz upwelling cell."**

*Lines 502-504. Here you should provide some hypotheses to explain why ladderane IPLs were not detected throughout the water column, whereas ladderane FA concentration increased with depth. Where are ladderane FAs derived from? What about potential influence of lateral transport?* **Line 502-504 is amended to:**

**"…column community. Possibly, BHT-x and ladderane FAs at this station were laterally transported from more southern shelf sites (Mollenhauer et al., 2007; Blumenberg et al., 2010)."**

*Please check the salinity scale in Fig. 5c.* **We thank the reviewer for spotting this error in the number of decimal places, which is now increased to two (giving 35.20, 35.25 and 35.30 as scale intervals).**

*Line 515. Similarly here, the relationship between ladderane IPLs and FAs should be better explained. Despite high abundance of ladderane IPLs, high abundance of ladderane FAs is not observed. This temporal offset should be discussed in more detail than just the sentence in lines 515-517.*

**Differences in degradation rate between ladderane IPLs and ladderane FAs are now described in more detail in section 4.1.3. In addition, this specific sentence is amended to:**

**"The ladderane FA and BHT-x concentrations were slightly lower then observed in February at 85 mbss, which may indicate that particulate material sank to the sea-floor, was degraded, or was transported elsewhere prior to the occurrence of an established anammox community in March."**

*Line 537. What do you mean by "well-known PCR biases"? This is unclear for the non-specialists.* **Amended to: "…Even so, it cannot be excluded that well-known PCR biases might also have led to a low coverage of *Ca.* Scalindua spp. reads. Unequal amplification efficiency of PCR products could result in the preferential amplification of certain 16S rRNA genes, whilst others might be inhibited for amplification (e.g. Pinto & Raskin, 2012).**

*Ca.* Scalindua...”

*Fig. 7. The numbering in the caption and in the figure is not consistent.* **Amended.**

*Lines 563-566. This threshold should be tested in other sites, this could be mentioned.*

**We agree with the reviewer that comparing the BHT-x ratios observed in the BUS with BHT-x ratios observed at other sites would be a valuable contribution to the manuscript. In the revised manuscript, we have included a new figure (see figure 9), which shows the relationship between oxygen concentrations and the BHT-x ratio in water columns from the Sáenz et al., (2011) and Matys et al., (2017) studies, as well as our own study. In order to compare our datasets, oxygen concentrations of our own study were converted to µmol kg$^{-1}$. Based on the data presented in this figure and the discussion in the rebuttal to referee 1, we have decreased the number of significant digits in the BHT-*x* ratio threshold value, to allow for a less constrained threshold (i.e. from 0.18 to 0.2). Also, the combination of our own dataset with those of by Sáenz et al. (2011) and Matys et al., (2017), shows that when [O$_2$] is <50 µmol kg$^{-1}$, the BHT-x ratio is ≥ 0.2 (except 1 sample in the Cariaco Basin; see figure 1). Considering the large variety in marine settings (including four different upwelling regions and one restricted anoxic basin) and in methodologies (Soxhlet versus modified Bligh & Dyer; UHPLC-APCI-MS versus UHPLC-ESI-MS), we believe that a BHT-x ratio of 0.2 provides a robust threshold to estimate lox oxygen conditions (<50 µmol kg$^{-1}$) in sedimentary records of various marine settings, including upwelling regions. See point 1 in comments posted by referee #1 for exact text amendments.**

**References:**

[revised manuscript text omitted]